# A salivary GMC oxidoreductase of *Manduca sexta* re-arranges the green leaf volatile profile of its host plant

Yu-Hsien Lin [1], Juliette J. M. Silven [1], Nicky Wybouw [2], Richard A. Fandino [3,7], Henk L. Dekker [4], Heiko Vogel [5], Yueh-Lung Wu [6], Chris de Koster [4,9], Ewald Große-Wilde [3,8], Michel A. Haring [1], Robert C. Schuurink [1] & Silke Allmann [1] ✉

Green leaf volatiles (GLVs) are short-chain oxylipins that are emitted from plants in response to stress. Previous studies have shown that oral secretions (OS) of the tobacco hornworm *Manduca sexta*, introduced into plant wounds during feeding, catalyze the re-arrangement of GLVs from *Z*-3- to *E*-2-isomers. This change in the volatile signal however is bittersweet for the insect as it can be used by their natural enemies, as a prey location cue. Here we show that (3*Z*):(2*E*)-hexenal isomerase (*Hi-1*) in *M. sexta's* OS catalyzes the conversion of the GLV *Z*-3-hexenal to *E*-2-hexenal. *Hi-1* mutants that were raised on a GLV-free diet showed developmental disorders, indicating that Hi-1 also metabolizes other substrates important for the insect's development. Phylogenetic analysis placed Hi-1 within the GMCβ-subfamily and showed that Hi-1 homologs from other lepidopterans could catalyze similar reactions. Our results indicate that *Hi-1* not only modulates the plant's GLV-bouquet but also functions in insect development.

Insect herbivores introduce small molecules into leaf wounds while feeding on their host plant. This has far-reaching consequences, not only for the plant itself but also for the attacking herbivore; plants are able to recognize the identity of their attacker and adapt their defense mechanisms accordingly[1–3]. Molecules, such as fatty-acid-derived metabolites, proteins, and peptides, that are present in the oral secretions (OS) of herbivores and transferred to the plant wound during feeding, play a major role in this[4,5]. Depending on their ability to either induce or suppress plant defense responses, these molecules are called elicitor or effector compounds, respectively[6].

One of the plant responses to herbivore attack is the release of volatile organic compounds (VOCs) that serve a multitude of functions. Some VOCs act as direct repellents, or even toxins to the herbivore, while others are used as air-borne signals for the herbivores' natural enemies and neighboring plants[7–9]. For predators and parasitoids, this volatile information helps to localize prey in proximity, while for nearby plants or adjacent leaves of the same plant, VOCs contribute to the preparations for imminent herbivore attack[10]. Green leaf volatiles (GLVs) belong to a class of VOCs that are immediately released by plants when damaged[11,12]. GLVs consist of six carbon (C6) aldehydes, alcohols, and esters, generated

[1]Green Life Sciences Research Cluster, Department of Plant Physiology, Swammerdam Institute for Life Sciences, University of Amsterdam, Amsterdam, Netherlands. [2]Terrestrial Ecology Unit, Department of Biology, Faculty of Sciences, Ghent University, Ghent, Belgium. [3]Department of Evolutionary Neuroethology, Max Planck Institute for Chemical Ecology, Jena, Germany. [4]Laboratory for Mass Spectrometry of Biomolecules, Swammerdam Institute for Life Sciences, University of Amsterdam, Amsterdam, Netherlands. [5]Department of Insect Symbiosis, Max Planck Institute for Chemical Ecology, Jena, Germany. [6]Department of Entomology, National Taiwan University, Taipei, Taiwan. [7]Present address: Department of Ecology & Evolutionary Biology, Cornell University, Ithaca, NY, US. [8]Present address: EXTEMIT-K, Faculty of Forestry and Wood Sciences, Czech University of Life Sciences, 16500 Prague, Czech Republic. [9]Deceased: Chris de Koster. ✉e-mail: S.Allmann@uva.nl

through the oxylipin pathway from C18-polyunsaturated fatty acids, and play roles in mediating both, direct and indirect defenses[13–16].

Research in recent years has shown that caterpillar feeding or the introduction of OS into plant wounds can have diverse effects on the release of GLVs from plants. A meta-analysis revealed that chewing herbivores primarily cause a suppression of the wound-induced emission of GLVs[12]. This result was rather surprising as it was long assumed that herbivory had no additional effect on the wound-induced release of GLVs, i.e., neither amplification nor suppression of the wound-induced GLV-burst[17–19]. By now various studies have shown that OS-derived compounds can suppress the release of GLVs[4,20], and some salivary enzymes causing this suppression have already been identified. *B. mori* larvae possess a fatty acid hydroperoxide dehydratase (BmFHD) in their OS that suppresses GLV production by scavenging a precursor of the GLV biosynthetic pathway[21]. Recently, FHD activity has also been identified in the OS of several other lepidopteran species[22]. Salivary glucose oxidase (GOX) of *H. zea* induces stomatal closure in tomato leaves, possibly by affecting plant $H_2O_2$ signaling, resulting in decreased emission rates of the GLVs *Z*-3-hexenol and *Z*-3-hexenyl acetate[23]. In addition, a heat-stable compound has been identified in the regurgitant of several caterpillars that suppresses GLV emission via an unknown, non-enzymatic mechanism[24].

Previous work has shown that insect OS can not only induce or suppress the wound-induced GLV release[12,25] but also modulate the profile of the released volatile compounds[22,26,27]. Earlier we discovered that the increased emission of *E*-2-isomers of GLVs, induced in *N. attenuata* by *M. sexta* feeding, strongly increased the foraging efficiency of the generalist predators *Geocoris* spp.[27] and decreased the oviposition rates of *M. sexta* conspecifics[26] in nature. Surprisingly, the conversion of the plant's *Z*-3-aldehydes to *E*-2-aldehydes was solely and directly due to isomerase activity in the herbivore's OS. Hence, we were left with the puzzling observation that *M. sexta* caterpillars are betraying themselves by making the plant's volatile bouquet more attractive to their natural enemy.

Here, we conducted a biochemical and proteomic analysis of *M. sexta* OS, to identify the first insect-derived compound that modulates the release of GLVs by re-arranging *Z*-3-hexenal (Z3AL) to *E*-2-hexenal (E2AL). In vitro application of recombinant protein of this (3*Z*):(2*E*)-hexenal isomerase (referred to as Hi-1) catalyzed the conversion of Z3AL to E2AL. Additionally, ectopic expression of *Hi-1* in transgenic *Arabidopsis thaliana* and *Nicotiana benthamiana* confirmed Hi activity by significantly increasing the E2AL emission in these plants. Mutagenesis of *Hi-1* by CRISPR/Cas9 not only confirmed that Hi-1 is necessary and sufficient for Hi activity in OS of *M. sexta*; we also observed significant growth disorders in *Hi-1* mutants indicating an important role in insect development. We investigated the evolutionary origin of *Hi-1* by a maximum-likelihood phylogenetic analysis using the GMC oxidoreductase multi-gene families of six lepidopteran species. Together with OS activity assays for this lepidopteran panel, we identified candidate *Hi-1* homologs from other lepidopteran species that could catalyze similar GLV conversions. Our study identified an insect-derived enzyme that acts as a "double-edged sword" by causing the betrayal of the insect to its own enemies[27], while at the same time influencing the insect's own development.

## Results

### *M. sexta* OS causes the aldehyde-specific re-arrangement of *Z*-3- to *E*-2-hexenal

Previous work revealed that *M. sexta* herbivory and the application of OS to plant wounds profoundly change the *Z*-3/*E*-2-ratio of plant-released GLVs[27]. In vitro experiments confirmed that a heat-labile compound, present in *M. sexta* OS, converts the aldehyde *Z*-3-hexenal (Z3AL) to *E*-2-hexenal (E2AL). However, it remains unclear whether the observed OS-induced increase in *E*-2-hexenol (E2OL) and *E*-2-hexenyl

acetate (E2oAc)[27] are merely the effects of elevated levels of the precursor compound E2AL, or whether the corresponding *Z*-3-GLVs, *Z*-3-hexenol (Z3OL) and *Z*-3-hexenyl acetate (Z3oAc) serve as substrates for the OS-derived proteinaceous compound as well (Fig. 1a). To test this, Z3AL (aldehyde), Z3OL (alcohol) or Z3oAc (ester) was incubated with OS from fifth instar *M. sexta* larvae and *Z*-3-/*E*-2-conversion was monitored by SPME-GC-qToF-MS (Fig. 1b). While *M. sexta* OS caused a clear conversion from Z3AL to E2AL, volatile profiles were unchanged when either Z3OL or Z3oAc was used as substrate. To test whether OS also catalyzes the reverse reaction at biologically relevant

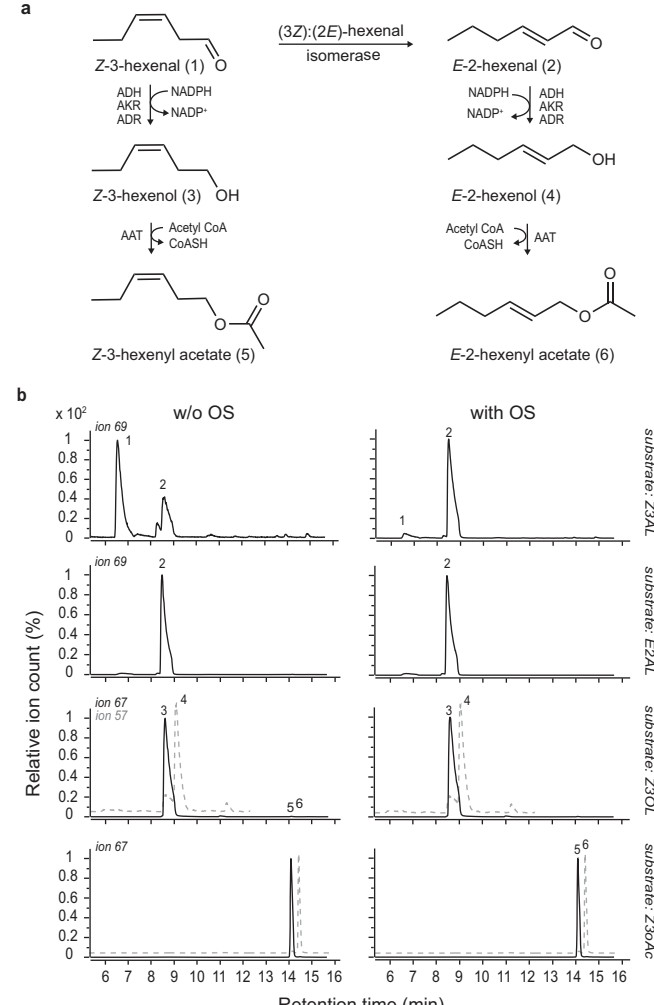

**Fig. 1 | Oral secretions (OS) of *Manduca sexta* catalyze the conversion of *Z*-3-hexenal to *E*-2-hexenal.** **a** Part of the biosynthetic pathway of green leaf volatiles (GLVs). The re-arrangement of *Z*-3-hexenal (Z3AL (1)) to *E*-2-hexenal (E2AL (2)) can either occur spontaneously or enzymatically by a (3*Z*):(2*E*)-hexenal isomerase. The (*Z*-3)/(*E*-2)-aldehydes can be further converted to their corresponding alcohols and esters. ADH, alcohol dehydrogenase; AKR, aldo-keto reductase; ADR, aldehyde reductase; AAT, alcohol acetyl transferase. **b** Representative chromatograms of headspace samples from *Z*-3-GLVs (Z3AL (1), *Z*-3-hexenol (Z3OL (3)) or *Z*-3-hexenyl acetate (Z3oAc (5)) and E2AL (2) that were either incubated with *M. sexta* OS from 5th instar larvae (with OS) or with a buffer as a control (w/o OS). Volatiles were identified using standard compounds. For simplification, the chromatograms of the standard solutions *E*-2-hexenol (E2OL (4)) and *E*-2-hexenyl acetate (E2oAc (6)), indicated by a grey, dotted line, are overlaid with the chromatograms in which Z3OL or Z3oAc were used as substrates. Ion 69 was used as qualifier ion for Z3AL and E2AL. Ion 67 was used as qualifier ion for Z3OL, Z3oAc and E2oAc and ion 57 was used as qualifier ion for E2OL. Headspace analysis for all samples and treatments was done in triplicates. **a**, **b** Volatile compounds in Fig. 1a are numbered and correspond to those shown in the chromatograms.

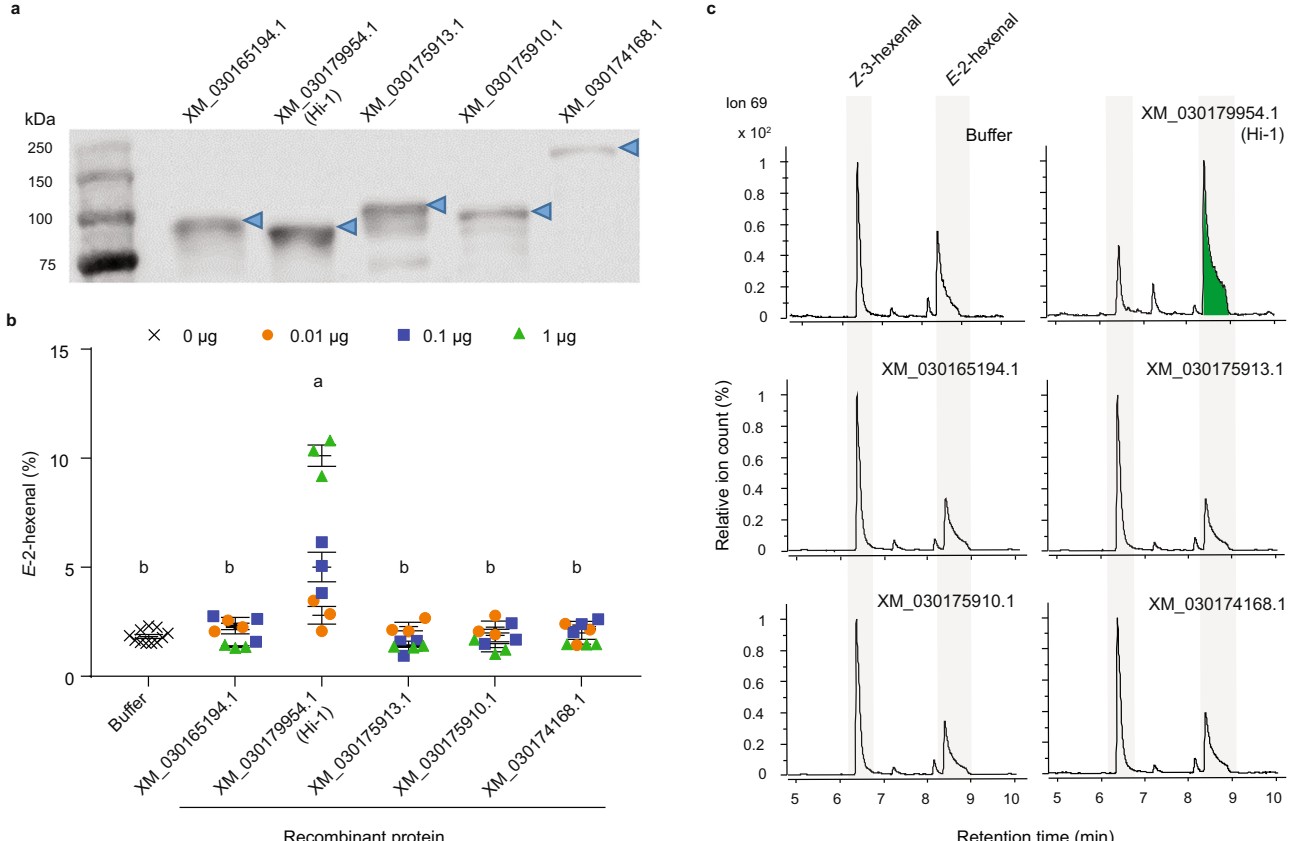

**Fig. 2 | Identification of a (3*Z*):(2*E*)-hexenal isomerase using purified recombinant proteins. a** Purified GST-tagged recombinant protein of five candidate proteins (XM_030165194.1, XM_030179954.1, XM_030175913.1, XM_030175910.1, and XM_030174168.1) were visualized by western blot. Blue arrows indicate the position of each recombinant protein. **b** Results of the SPME-guided in vitro assay for measuring (3*Z*):(2*E*)-hexenal isomerase activity. Three different protein quantities (0.01, 0.1, or 1 μg) were incubated with *Z*-3-hexenal (0.2 mM), and the proportion of *E*-2-hexenal (in percentages) emitted from total aldehydes (*Z*-3-hexenal + *E*-2-hexenal) was calculated. Elution buffer (Tris-HCl pH 8.0 with 10 mM GSH) was used as a control. One-way ANOVA was performed to determine significant differences between buffer ($n = 11$ biologically independent samples) control and 0.1 μg ($n = 3$ biologically independent samples) of recombinant proteins ($F_{5,20} = 20.24$, $p < 0.0001$) or between buffer and 1 μg of recombinant proteins ($F_{5,20} = 302.8$, $p < 0.0001$) followed by a Tukey HSD post-hoc analysis. Different letters indicate significant difference ($p < 0.05$) between buffer control and 0.1 μg or 1 μg recombinant proteins. Data are normalized using the corresponding response factors of each compound. **c** Representative extracted ion chromatograms (ion 69) as output of the SPME-guided assay with 1 μg purified protein. Peak areas are pre-processed data and are thus not normalized by response factors. Green highlight in Hi-1 indicates a clear conversion of *Z*-3-hexenal to *E*-2-hexenal. Error bars are presented as mean values ± SEM.

concentrations, we incubated E2AL with *M. sexta* OS and monitored the production of Z3AL. We were unable to detect the conversion of E2AL back to Z3AL. This indicates that the equilibrium concentration position of the hexenal isomerase lies far to the right and that the enzyme specifically uses Z3AL as a substrate from all available GLVs.

### Purification of a (3*Z*):(2*E*)-hexenal isomerase (Hi) from *M. sexta* OS

To identify the protein responsible for the rearrangement of Z3AL to E2AL, we partially purified hexenal isomerase activity from *M. sexta* OS. For this, we collected 2 mL of OS from *M. sexta* that had fed on wild tobacco plants and used the in vitro hexenal isomerase assay to drive the fractionation of biological activity. Crude OS was centrifuged, filtered, and subsequently fractionated by size exclusion chromatography (Supplementary Fig. 1 step 1). As previous experiments showed that activity was present in the retentate of 50 kDa cut-off filters the pooled active fractions (marked as fraction a) and two pooled non-active fractions (b and c) were concentrated with 50 kDa cut-off filters and separated and visualized on a 10% SDS-PAGE gel. We focused on three bands that were clearly enriched in the pooled active fraction

(Fig S1, step 3). The three gel-slices that each contained a single visible protein band were digested with trypsin and analyzed by LC-MS/MS. In total we identified seven proteinaceous candidates from the three bands (Supplementary Data 1) of which we selected five candidates for further analysis using a three peptide hits threshold. To test whether one of these candidates conferred Hi activity, we cloned the corresponding cDNAs and expressed their recombinant proteins in *E. coli* (Fig. 2a and Supplementary Fig. 2). SPME-GC-qToF-MS analysis showed that one candidate, XM_030179954 (hereinafter referred to as (3*Z*):(2*E*)-hexenal isomerase 1 (Hi-1), possessed Hi activity. When purified Hi-1 protein was added to the Z3AL substrate, significantly higher levels of E2AL were detected upon incubation when compared to the buffer control (Fig. 2b, c). This conversion was concentration-dependent, as higher amounts of Hi-1 protein increased the levels of E2AL (Fig. 2b and Supplementary Fig. 3). In contrast, the other four candidates (XM_030165194, XM_030175913, XM_030175910, and XM_030174168) did not reveal Hi activity with any protein concentration tested. Kinetic parameters, $K_m$, $k_{cat}$, and $k_{cat}/K_m$ of Hi-1 with Z3AL as substrate were estimated to be 1.27 mM, 121 sec⁻¹, and 115 sec⁻¹ mM⁻¹, respectively (Supplementary Table 1).

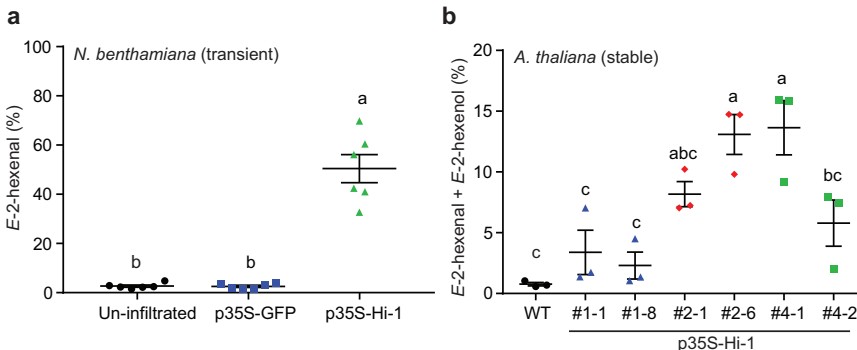

**Fig. 3 | Ectopic expression of *Manduca sexta*'s hexenal isomerase-1 (*Hi-1*) in *Nicotiana benthamiana* and *Arabidopsis thaliana* confers (3*Z*):(2*E*)-hexenal isomerase activity. a** Results of the SPME-guided in vitro assay for measuring hexenal isomerase activity in transgenic *N. benthamiana*. *Z*-3-hexenal (0.1 mM) was added to crude leaf extract of un-infiltrated plants or plants transiently expressing *Hi-1* or GFP as a control treatment, and samples were collected four days post infiltration. The proportion of *E*-2-hexenal (in percentages) emitted from the total aldehyde headspace (*Z*-3-hexenal + *E*-2-hexenal) was calculated. Different letters indicate significant differences ($p < 0.05$) between groups (One-way ANOVA followed by Tukey HSD post-hoc analysis, $F_{2,15} = 69.69$, $p < 0.0001$, $n = 6$ biologically

independent samples). **b** Measurement of (3*Z*):(2*E*)-hexenal isomerase activity in transgenic *Arabidopsis* (Col-0) T3-lines stably expressing *Hi-1*. Crude extracts of whole shoots from 5-weeks old plants were added to *Z*-3-hexenal (0.1 mM) to measure isomerase activity. Since the aldehyde substrate was partially converted into its alcohol by plant endogenous proteins, the hexenal isomerase activity was determined by calculating the proportion of *E*-2-hexenal + *E*-2-hexenol emitted from the total aldehyde + alcohol headspace. Different letters indicate significant differences ($p < 0.05$) between groups (One-way ANOVA followed by Tukey HSD post-hoc analysis, $F_{6,14} = 10.84$, $p = 0.0001$, $n = 3$ biologically independent samples). Error bars are presented as mean values ± SEM.

## Ectopic expression of *Hi-1* in *Nicotiana benthamiana* and *A. thaliana* confers Hi activity

To further confirm that *M. sexta* Hi-1 catalyzed the Hi reaction *in planta*, we ectopically expressed *Hi-1* in *N. benthamiana* and *A. thaliana*, two plant species that do not possess endogenous Hi activity[28,29]. The crude leaf extracts of *N. benthamiana* transiently expressing *Hi-1* revealed significantly higher Hi activity in comparison to the crude extracts of un-infiltrated and GFP expressing plants (Fig. 3a). In addition, crude leaf extracts from stable *A. thaliana Hi-1* overexpression lines showed clear Hi activity (Fig. 3b), especially in the independent lines #2–6 and #4–1 which also presented highest *Hi-1* transcript levels (Supplementary Fig. 4). The two lines 1 (#1–1, #1–8) with lowest *Hi-1* expression levels as well as wild-type Col-0 contained significantly lower or absent Hi activities.

## *Hi-1* re-arranges various *Z*-3-aldehydes

From research on plant hexenal isomerases we know that *Z*-3-aldehydes of different chain lengths can be suitable substrates[28,29]. To test whether this also holds true for insect-derived hexenal isomerases we assayed recombinant *M. sexta* Hi-1 for its ability to convert either *Z*-3-octenal or *Z*-3-nonenal to *E*-2-octenal or *E*-2-nonenal, respectively. Our SPME-guided analysis showed that recombinant *M. sexta* Hi-1 (0.1 μg) was able to convert both *Z*-3-aldehydes to the respective *E*-2-aldehydes (Supplementary Fig. 5).

## *Hi-1*, a salivary (labial) gland-specific gene, is highly expressed at early and late larval stages of *M. sexta*

To determine whether the transcript levels of *Hi-1* change throughout the life cycle of *M. sexta*, we compared gene expression of *Hi-1* across different developmental stages. Expression of *Hi-1* was high in neonates and first instar larvae (24 hours post hatching), significantly dropped in second and third instars, and increased again in fourth and fifth instars (Fig. 4a). The transcript levels of *Hi-1* were low at embryonic, pupal, and adult stages. Hi activity of OS from different instars revealed a similar pattern as observed for the *Hi-1* transcriptional profile (Fig. 4b, c). OS from the first, fourth, and fifth instars displayed high Hi activity while no clear activity was found in the OS of the second and third instars. To validate the low levels of activity in OS of second and third instar larvae we increased the amount of OS used for the in vitro

assay tenfold. However, conversion rates from Z3AL to E2AL still did not significantly differ from control treatments (Supplementary Fig. 6a, b).

To determine whether transcript levels of *Hi-1* display tissue specificity, we compared *Hi-1* expression across seven tissues including brain plus ganglia (BR), foregut (FG), midgut (MG), hindgut (HG), malpighian tubules (MT), fat body (FB) and salivary (labial) glands (SG) of fourth instar larvae. The qPCR result showed that *Hi-1* is exclusively and highly expressed in the salivary glands (Fig. 4d), indicating that Hi-1 is a salivary-specific protein. We thus conclude that Hi-1 is highly active at early and late larval stages and specifically expressed in salivary (labial) glands.

## Targeted mutagenesis of *Hi-1* by CRISPR/Cas9 and off-target assessment on gene duplicate (*Hi-like*)

Analysis of the Hi-1 amino acid sequence by InterPro (EMBL-EBI) uncovered a functional domain (IPR012132) that indicated that Hi-1 belongs to the glucose-methanol-choline (GMC) oxidoreductase family. Genome mining via tBLASTn- and BLASTp-searches identified a close paralog to *Hi-1*, XM_030179956, hereinafter referred to as *Hi-like*. *Hi-like* and *Hi-1* were linked and exhibited 85% coding sequence identity (Supplementary Fig. 7)[30]. However, recombinant protein of Hi-like (Fig. 5a, Supplementary Fig. 8) was unable to convert Z3AL to E2AL, showing that this close paralog did not possess any Hi activity (Fig. 5b, Supplementary Fig. 9). Additionally, in contrast to *Hi-1*, *Hi-like* was not specifically expressed in the salivary (labial) glands, but instead showed highest transcript levels in the midgut (Fig. 5c).

We applied CRISPR/Cas9 to generate a *Hi-1* null mutation in *M. sexta*. We designed a gRNA1 that targeted exon 4 of *Hi-1* and caused a 160 bp deletion in a knockout mutant line. This deletion led to a frameshift that resulted in the introduction of a premature stop codon located 82 bp downstream of the deletion site (Fig. 5d). Sanger sequencing of the *Hi-like* gene in wildtype (*Hi-1*[+]) and *Hi-1* knockout (*Hi-1*[-]) strains indicated that there was no off-target deletion on its gene paralog, *Hi-like*, even though 78% of the gRNA1 sequence (18 out of 23 nucleotides) was homologous to the last exon of *Hi-like* (Fig. 5e). We concluded that CRISPR/Cas9 successfully generated a *Hi-1* deletion without causing an off-target effect on the paralog, *Hi-like*.

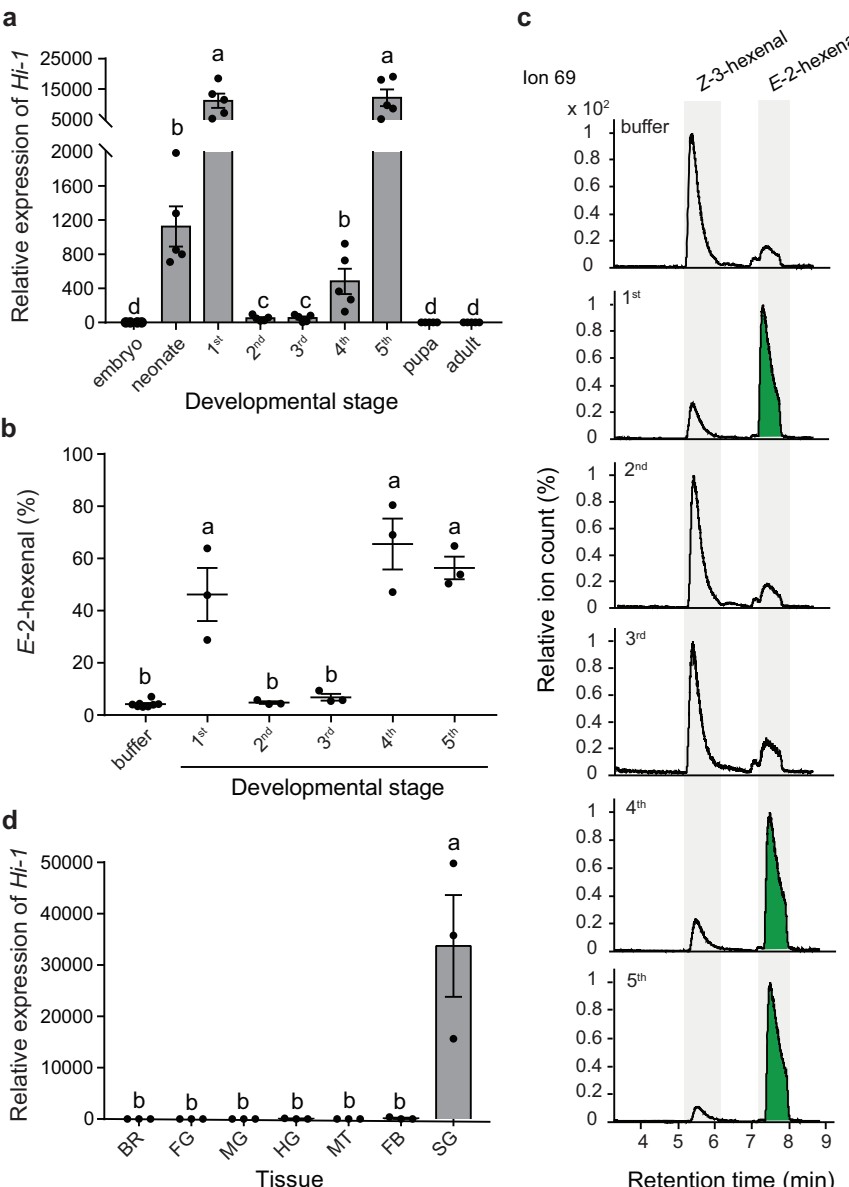

**Fig. 4 | _Hi-1_ acts as a salivary (labial) gland-specific gene and is highly expressed at early and late larval stages of _M. sexta_. a** Relative transcript abundance of _Hi-1_ throughout different developmental stages. Expression levels of each stage are relative to the expression level in the embryo. Different letters indicate significant differences ($p < 0.05$) between developmental stages (One-way ANOVA analysis with log2 transformed data followed by Tukey HSD post-hoc analysis, $F_{8,35} = 152.3$, $p < 0.0001$, $n = 5$ biologically independent samples). **b** (3Z):(2E)-hexenal isomerase activity of oral secretions from larvae at different developmental stages. Equal amounts of total protein (0.5 μg) per sample ($n = 3$ biologically independent samples) or buffer control ($n = 8$ biologically independent samples) were incubated with Z-3-hexenal (1 mM), and a SPME-guided in vitro assay was performed. The proportion of E-2-hexenal (in percentages) emitted from the total aldehyde

headspace (Z-3-hexenal + E-2-hexenal) was calculated. Different letters indicate significant differences ($p < 0.05$) between tissues (One-way ANOVA followed by Tukey HSD post-hoc analysis, $F_{5,17} = 38.21$, $p < 0.0001$). **c** Representative extracted ion chromatograms (ion 69) as output of the SPME-guided assay. Green highlight of the E-2-hexenal peak indicates a clear conversion of Z-3-hexenal to E-2-hexenal. **d** Relative transcript abundance of _Hi-1_ in different organ tissues including brain plus ganglia (BR), foregut (FG), midgut (MG), hindgut (HG), Malpighian tubules (MT), fat body (FB) and salivary glands (SG) of fourth instar larvae (expression relative to BR). Different letters indicate significant differences ($p < 0.05$) between tissues (One-way ANOVA analysis with log2 transformed data followed by Tukey HSD post-hoc analysis, $F_{6,14} = 10.05$, $p = 0.0002$, $n = 3$ biologically independent samples). Error bars are presented as mean values ± SEM.

### OS of _Hi-1_ knockout mutant lost the ability to convert Z3AL to E2AL

To determine whether the _Hi-1_ mutation resulted in the loss of Hi activity in _M. sexta_, we compared Hi activity of OS from 5th instar _Hi-1_ homozygous mutants (_Hi-1⁻_), heterozygous mutants (_Hi-1⁺/⁻_) and wild-type (_Hi-1⁺_). The Hi activity was abolished in _Hi-1⁻_ OS even after increasing the amount of purified protein ten times (Fig. 6a, b). We also did not observe a significant change of Hi activity after heat treatment of _Hi-1⁻_ OS, in contrast to heat-treated _Hi-1⁺_ OS which lost Hi activity

completely (Supplementary Fig. 10a, b). Moreover, _Hi-1⁺/⁻_ OS showed decreased Hi activity in comparison to _Hi-1⁺_ OS (Fig. 6a, b), indicating that mutating a single _Hi_ allele was sufficient to influence the OS-derived Hi activity. To test whether _Hi-1⁻_ OS had lost its ability to change the GLV profile _in planta_, we compared the early GLV-bouquet of mechanically wounded _N. attenuata_ leaf disks that were either treated with water or OS collected from homozygous mutants or wild-type caterpillars. We observed that both E2AL and E2OL emissions significantly increased when _Hi-1⁺_ OS was applied to wounded leaf

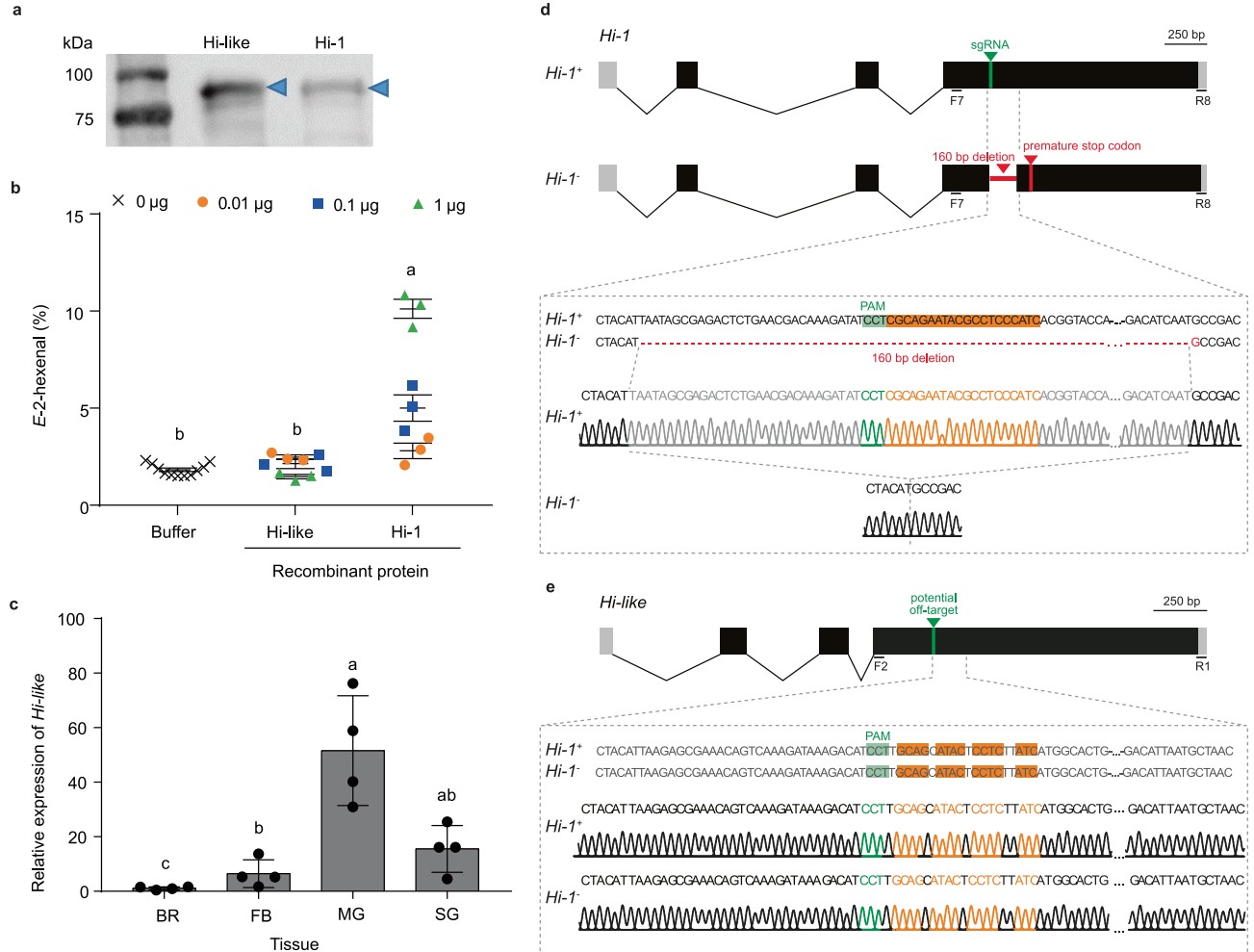

**Fig. 5 | CRISPR/Cas9 mutagenesis of *Hi-1* without off-target deletion of the paralog *Hi-like* which lacks (3*Z*):(2*E*)-hexenal isomerase activity. a** Purified GST-tagged recombinant proteins of Hi-like and Hi-1 were visualized on western blot. Blue arrows indicate the position of each recombinant protein on the gel. **b** Results of the SPME-guided in vitro assay for measuring (3*Z*):(2*E*)-hexenal isomerase activity. Each protein with three different concentrations (0.01, 0.1, or 1 μg) was incubated with *Z*-3-hexenal (0.2 mM), and the proportion of *E*-2-hexenal from the total aldehyde headspace (*Z*-3-hexenal + *E*-2-hexenal) was determined. Elution buffer (Tris-HCl pH 8.0 with 10 mM GSH) was used as a control. One-way ANOVA followed by a Tukey HSD post-hoc analysis was performed to determine significant differences between buffer control (*n* = 11 biologically independent samples) and samples containing 1 μg of recombinant protein (*n* = 3) ($F_{2,14}$ = 504.7, *p* < 0.0001). Different letters indicate significant differences (*p* < 0.05) between control and recombinant proteins. **c** Tissue-specific expression of *Hi-like* in brain plus ganglia (BR), fat body (FB), midgut (MG), and salivary (labial) glands (SG) of fourth instar larvae (expression relative to BR). Different letters indicate significant differences (*p* < 0.05) between tissues (One-way ANOVA analysis with log2 transformed data followed by Tukey HSD post-hoc analysis, $F_{3,12}$ = 24.11, *p* < 0.0001, *n* = 4 biologically independent samples). **d** The gene structure of wildtype *Hi-1* (*Hi-1⁺*) was accessed through NCBI (Gene ID: 115451596). The sgRNA targeted exon four of *Hi-1* and generated a 160 bp deletion in the *Hi-1* mutant (*Hi-1⁻*) resulting in a frameshift and premature stop codon located 82 bp downstream of the deletion side. Sanger sequencing profiles show genomic *Hi-1* sequences of *Hi-1⁺* or *Hi-1⁻*, green and orange colors represent PAM sequence and sgRNA target sites, respectively. **e** The gene structure of *Hi-like* was accessed through NCBI (Gene ID: 115451597), and potential off-target site of *Hi-1* sgRNA was highlighted at the fourth exon of *Hi-like*. Sanger sequencing profile shows genomic *Hi-like* sequences of *Hi-1⁺* or *Hi-1⁻*; no off-target deletion was found in both wildtype and *Hi-1* mutant strains. Green and orange sequences indicate overlapping PAM sequences and *Hi-1* sgRNA target sites. Error bars are presented as mean values ± SEM.

disks. However, GLV-profiles of leaf discs that were either treated with water or *Hi-1⁻* OS did not differ from each other (Fig. 6c).

### *Hi-1* contributes to larval development and metamorphosis in *M. sexta*

In previous studies we have shown that the conversion from Z3AL to E2AL can be both, detrimental[27] and beneficial[26] for *M. sexta*. To determine whether Hi-1 plays other essential physiological roles, we compared the larval and adult developmental profiles and adult morphologies between *Hi-1⁺* and *Hi-1⁻* for three consecutive generations. Larvae were fed on artificial diet to focus on the physiological consequences of *Hi-1* mutation that were independent of the host plant and thus from GLV-isomerization. We observed that, for both female and male *Hi-1⁻*, the developmental time from neonate to adult

stage was extended compared to *Hi-1⁺* (Fig. 7a). While in males the pupal weight of *Hi-1⁻* was lower than that of *Hi-1⁺* we did not observe a weight difference in female pupae (Fig. 7b). Although the adult weights of both sexes did not differ between mutants and WT (Fig. 7c), we observed higher amounts of morphological deformations (including proboscis misfolding, loss of thoracic scales and non-expanded wings) among newly emerged *Hi-1⁻* adults (Fig. 7d). The number of deformations in adults of the *Hi-1⁻* colony was 2 to 3 times higher than that of the *Hi-1⁺* colony (Fig. 7e). Furthermore, the number of individuals with multiple deformations, especially double deformations, clearly increased in the *Hi-1⁻* individuals in both male and female (Fig. 7f). We observed the same developmental delays and morphological deformations in adult mutants when kept at slightly lower temperature (24 °C; Supplementary Fig. 11). We thus conclude that *Hi-1* most likely

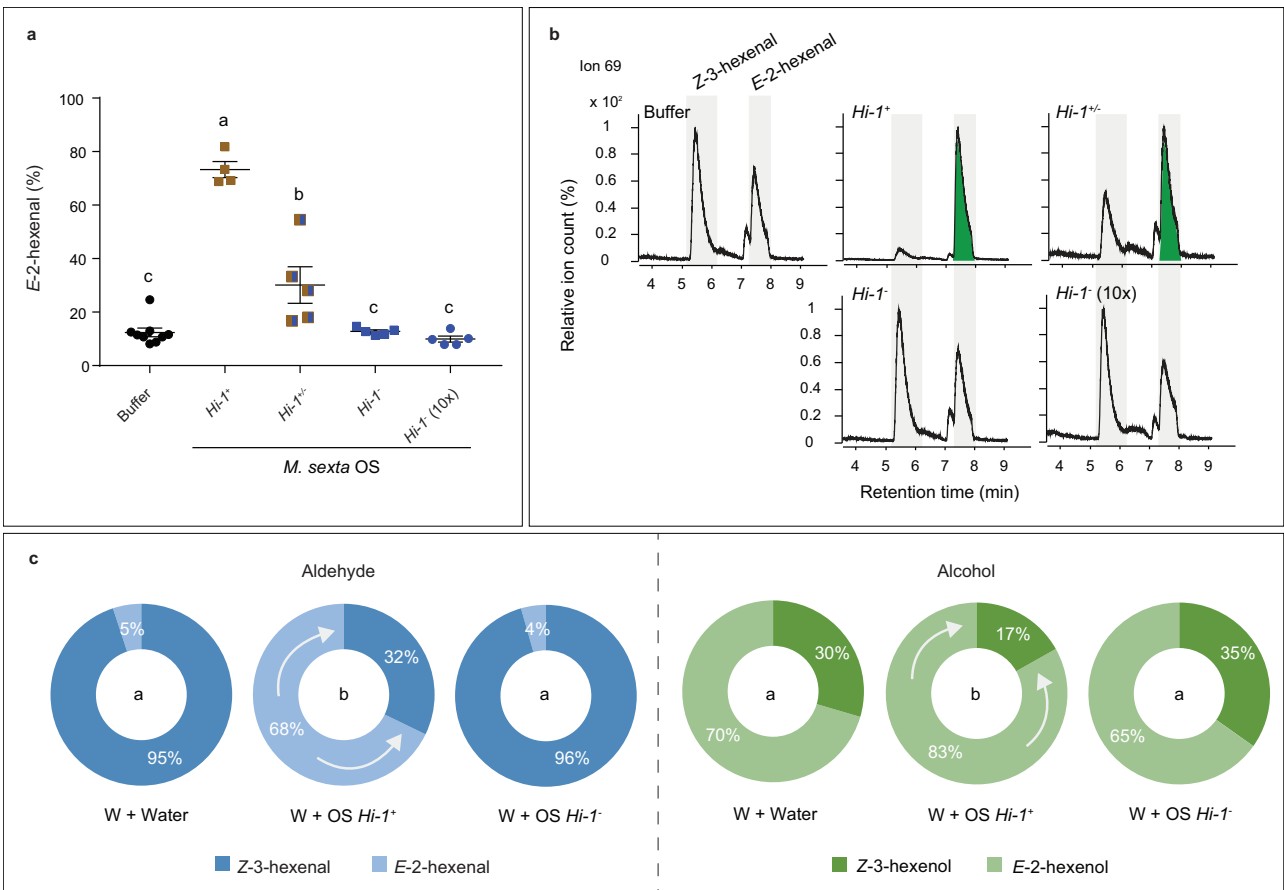

**Fig. 6 | Oral secretions (OS) of *Hi-1* mutant have no (3*Z*):(2*E*)-hexenal isomerase activity. a** Hexenal isomerase activity of OS from 5th instar wild-type (*Hi-1⁺*), heterozygous mutant (*Hi-1⁺/⁻*) and homozygous mutant (*Hi-1⁻*) larvae was determined by SPME-guided in vitro assay. Equal amounts (0.33 µg) of total protein per sample or a tenfold amount (3.3 µg) of homozygous mutant OS (labeled as *Hi-1⁻*(10x)) were incubated with *Z*-3-hexenal (0.2 mM). Buffer was used as control. The proportion of *E*-2-hexenal (in percentages) emitted from the total aldehyde headspace (*Z*-3-hexenal + *E*-2-hexenal) was calculated. Different letters indicate significant differences (*p* < 0.05) between tissues (One-way ANOVA followed by Tukey HSD post-hoc analysis, $F_{4,23}$ = 57.97, *p* < 0.0001, *n* = 9 for buffer, *n* = 4 for ***Hi-1⁺***, *n* = 5 for *Hi-1⁺/⁻*, *Hi-1⁻*, and *Hi-1⁻* (10x) biologically independent samples). **b** Representative extracted

ion chromatograms (ion 69) as an output of the SPME-guided assay. The green highlight in *E*-2-hexenal peak area indicates a clear conversion of *Z*-3-hexenal to *E*-2-hexenal. **c** Emission of GLVs from leaf discs of *N. attenuata*. Leaf discs (2.4 mm diameter) were mechanically wounded and treated with 10 µL of Milli-Q water (w + water; *n* = 4 biologically independent samples), OS (10 µg total protein) from *Hi-1⁺* (w + OS *Hi-1⁺*; *n* = 3 biologically independent samples) or from *Hi-1⁻* (w + OS *Hi-1⁻*; *n* = 3 biologically independent samples). The GLV composition was determined by SPME-GC-qToF-MS. One-way ANOVA was performed to identify significant differences of E2AL ($F_{2,7}$ = 32.44, *p* = 0.0003) or E2OL ($F_{2,7}$ = 9.887, *p* = 0.0091) between treatments. Different letters indicate significant differences (*p* < 0.05) by Tukey HSD post-hoc analysis. Error bars are presented as mean values ± SEM.

plays a role in the development of *M. sexta* as both, male developmental time till pupation and morphology of adults is affected in individuals that have lost larval Hi-1 activity.

**Hexenal isomerase activity is widespread in Lepidoptera and shows a strong association with the GMCβ subfamily**
We identified a total of 257 GMC genes in the annotated genomes of five lepidopteran species (*M. sexta, Pieris rapae, Chloridea virescens, Spodoptera litura*, and *Danaus plexippus*) by sequence homology detection. We constructed a maximum-likelihood phylogram incorporating 124 reference GMC genes of various lineages, including the domestic silk moth *Bombyx mori*[31] (Fig. 8a). Consistent with previous studies, the GMC genes of leaf beetles that code for salicyl alcohol oxidases (SAOs) and 8-hydroxygeraniol oxidoreductases (HGOs) were placed in different GMC subfamilies (GMCι and GMCκ, respectively)[32–34]. Phylogenetic reconstruction uncovered that *Hi-1* of *M. sexta* and its gene duplicate *Hi-like* are embedded within a well-supported group of the GMCβ subfamily that consisted exclusively of 60 lepidopteran GMCs (indicated by a green dashed circumferential line in Fig. 8a and focal tree in Fig. 8b). *Hi-like* and *Hi-1* clustered together with high node support, further suggesting that the two

paralogs are the result of a recent, species-specific gene duplication event. One of our initial Hi-candidates with no Hi-activity, that we identified from proteomic analysis (XM_030165194) (Fig. 2 and Supplementary Data 1) was also placed in the GMCβ subfamily, but distinct from the Hi-1 subclade (Fig. 8a, highlighted with "s"). In contrast to the other species of our lepidopteran panel, *P. rapae* only carried a single gene within this group. The single *P. rapae* homolog did not cluster strongly with any other homolog of this GMCβ subclade (Fig. 8b).

To gain more insight into the evolutionary origin(s) of hexenal isomerase activity in Lepidoptera, Hi-activity was tested in the OS (with 4 µg total protein) of our lepidopteran species panel, *C. virescens, S. litura, D. plexippus, B. mori* and *P. rapae* (Fig. 8c and Supplementary Fig. 12a). The chromatograms showed clear conversion to E2AL with OS from *C. virescens, S. litura, D. plexippus* and *B. mori*, species that possess numerous genes in the distinct subclade of GMCβ (Fig. 8b). In contrast, OS from *P. rapae* was unable to convert Z3AL to E2AL. To verify that the Hi activity in the OS of the caterpillars was not derived from ingested plant leaf material and thus originated from plant-derived hexenal isomerases, we examined Hi activity of leaves from milkweed (*Asclepias incarnata*), mulberry (*Morus alba*) and *Arabidopsis thaliana* which were used to feed *D. plexippus, B. mori* and *P.*

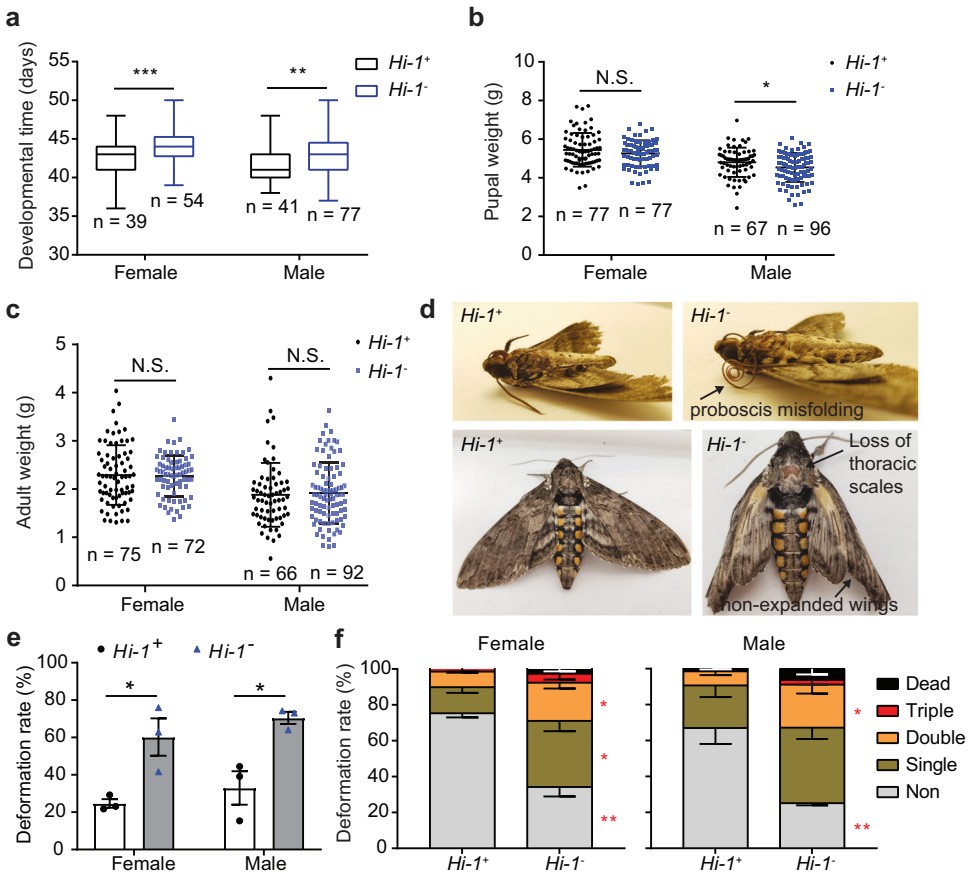

**Fig. 7 | Mutation of *Hi-1* causes developmental disorders and adult deformation in *M. sexta* reared on artificial diet. a** The developmental time was recorded from neonate to newly emerged adult stage. Two-tailed t-tests with Welch's correction were performed to compare between wildtype (*Hi-1⁺*) and mutant (*Hi-1⁻*) female (t = 3.58, df = 80.42, p = 0.0006) or male (t = 3.29, df = 88.37, p = 0.0014). For the box plot, the lower and upper hinges correspond to the 25th and 75th percentiles with the median as a line within interquartile range (IQR). The whiskers extend to the smallest and largest values within 1.5 times the IQR. *n* represents biologically independent samples. **b** Pupal weight was measured from the pupae formed 6-7 days after the wandering stage. Two-tailed t-test with Welch's correction was performed to compare between *Hi-1⁺* and *Hi-1⁻* female (t = 1.45, df = 143.5, p = 0.1483) or male (t = 2.2, df = 141.7, p = 0.0287). *n* represents biologically independent samples. **c** Adult weight was measured from the newly emerged adults (within 24 hours). Two-tailed t-test with Welch's correction was performed comparing between *Hi-1⁺* and *Hi-1⁻* female (t = 0.27, df = 131.6, p = 0.7862) or male (t = 0.36, df = 135.6,

p = 0.7234). *n* represents biologically independent samples. **d** A higher frequency of morphological deformations, including proboscis misfolding, loss of thoracic scales, and non-expanded wings was identified in *Hi-1⁻* adults. **e** Proportions of deformed adults were recorded from three separate generations (n = 3 biologically independent samples). Two-tailed t-test was performed to compare between *Hi-1⁺* and *Hi-1⁻* female (t = 3.46, df = 4, p = 0.0259) or male (t = 3.95, df = 4, p = 0.0168). **f** Percentages of adults that displayed non-, single, or multiple deformation, or died while in pupal stage or eclosion (n = 3 biologically independent samples). Two-tailed t-test was performed to compare between *Hi-1⁺* and *Hi-1⁻* female (df = 4; Non, t = 7.11, p = 0.0021; Single, t = 3.41, p = 0.00268; Double, t = 3.81, p = 0.0189; Triple, t = 1, p = 0.3703; Dead, t = 1.99, p = 0.1175) or male (df = 4; Non, t = 4.65, p = 0.0097; Single, t = 2, p = 0.1105; Double, t = 2.97, p = 0.0410; Triple, t = 1, p = 0.3739; Dead, t = 1.38, p = 0.2389). Significant differences of all graphs within this figure are labeled as follows: *p < 0.05; **p < 0.01; **p < 0.001; N.S., not significant. Error bars are presented as mean values ± SEM.

*rapae*, respectively. The application of any leaf crude extract (using protein concentrations equal to the OS-samples i.e. 4 μg total protein) did not cause an increased conversion of the substrate Z3AL to E2AL compared to the buffer control (Supplementary Fig. 12b) even after increasing the amount of leaf crude extract tenfold (40 μg, Supplementary Fig. 12c). This suggests that the identified Hi activities in the OS of plant-fed lepidopteran species were not derived from plants but were catalyzed by insect salivary proteins.

We observed clear Hi activity in the OS of *B. mori*, a moth that, like *M. sexta*, is placed within the Bombycoidea superfamily and is the closest relative to *M. sexta* within our caterpillar panel. As this result is in contrast to an earlier study from Jones, et al.[22] in which the authors were unable to detect such activity in the OS of *B. mori* larvae, we searched for putative homologs of MsHi-1 in the *B. mori* genome. Our phylogenetic analysis revealed the presence of two homologs in *B. mori* (BGIBMGA000068, BGIBMGA000158), both of which are expressed in the silk (labial) glands (Supplementary Fig. 13; SilkDB 3.0, Lu, et al.[35]), while only *BGIBMGA000158*

(hereafter referred as BmHi) is exclusively expressed in silk glands of fourth and fifth instar larvae – those developmental stages that possess high Hi-activity in *M. sexta*. BmHi recombinant protein (Supplementary Fig. 14a, b) converted Z3AL to E2AL in vitro (Supplementary Fig. 13c), though its activity seemed to be slightly lower than that of MsHi-1 recombinant protein (Fig. 2c). Thus, while we cannot fully explain the difference in results reported between the two studies, our data clearly shows that *B. mori* possess an enzyme that is able to convert Z3AL to E2AL in vivo (Fig. 8c) as well as in vitro (Supplementary Fig. 13c).

Together, the results of the phylogenetic reconstruction and the OS activity assays show that none of the *P. rapae* GMC homologs catalyze a hexenal isomerase activity, including the homolog that is weakly associated with *M. sexta* Hi-1. Based on these observations, we propose to narrow down candidate GMC genes that code for hexenal isomerase activity to a more internal node that contained 53 lepidopteran GMCs (indicated by a blue dashed circumferential line in Fig. 8a and blue star in Fig. 8b).

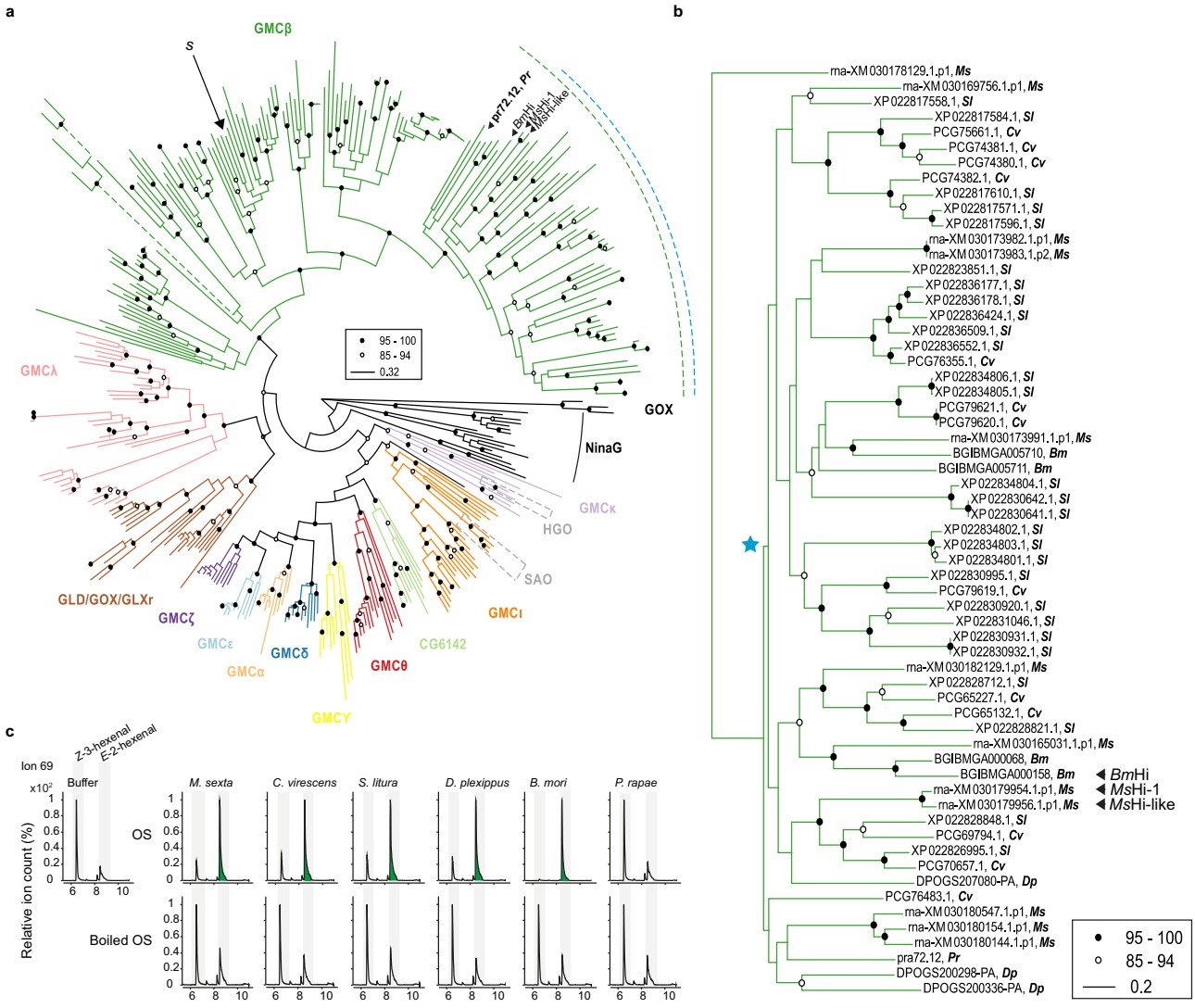

**Fig. 8 | Hexenal isomerase activity in oral secretions is pervasive in Lepidoptera and is associated with a specific cluster within the GMCβ subfamily. a** A maximum-likelihood tree for the GMC multi-gene family was generated using 381 GMC protein sequences and rooted using fungal GOX sequences. The initial candidate XM_030165194, which was present in the active fraction of the partially purified *M. sexta* OS and that showed no Hi activity (Fig. 2) is highlighted with "s". The leaf beetle GMCs that have been functionally characterized as HGOs or SAOs are delineated by grey dashed lines. The lengths of dashed branches have been reduced and are not representative for phylogenetic distance. The well-supported clade that contains MsHi-1 is indicated by a green dashed circumferential line. Our final set of candidate GMC genes that could catalyze Hi activity is indicated by a blue dashed circumferential line. Accession numbers can be found in the Supplementary Data 2. **b** Focal tree of the GMC proteins that were within the green dashed line of Fig. 8a. The blue star indicates the subclade of candidate Hi homologs based on both phylogenetic and Hi activity evidence (blue dashed line Fig. 8a). The tree was generated using 60 GMCβ protein sequences and rooted using *M. sexta* XM_030178129.1. The lepidopteran species *B. mori, C. virescens, D. plexippus, M. sexta, P. rapae, S. litura* are abbreviated as *Bm, Cv, Dp, Ms, Pr,* and *Sl*, respectively. For both phylogenetic trees, ultrafast bootstrap support values between 85 and 94% are indicated by empty black circles, whereas ultrafast bootstrap support values of 95% or higher are indicated by full black circles. **c** Representative extracted ion chromatograms (ion 69) of the SPME-guided assay to determine Hi activity in OS from different lepidopterans. OS, containing equal amounts of total protein, collected from 4th to 5th instar larvae were incubated with Z-3-hexenal, and hexenal isomerase activity was determined by the SPME-guided in vitro assay. The proportion of *E*-2-hexenal emitted from the total aldehyde headspace (Z-3-hexenal + *E*-2-hexenal) was calculated. Statistical analysis is presented in Supplementary Fig. 12a. Green highlight in the *E*-2-hexenal peak indicates clear conversion of Z-3-hexenal to *E*-2-hexenal.

## Discussion

Insect OS can modulate the release of herbivore-induced plant volatiles, thereby affecting the plant's capacity to communicate with its environment[36,37]. While certain salivary compounds cause an increase in the release[38,39] and/or initiate *de-novo* synthesis of herbivore-specific volatiles[40], other salivary components suppress the amount of volatiles emitted by plants[21,23,41]. In this study, we identified and characterized the first salivary enzyme that neither induces nor suppresses the release of plant volatiles, but rather modifies the composition of those plant volatiles that are emitted very early upon herbivory, the so-called green leaf volatiles (GLVs). Here, we identified a salivary GMC oxidoreductase from the tobacco hornworm, *M. sexta*, as a (3*Z*):(2*E*)-hexenal isomerase 1 (Hi-1) that catalyzes the re-arrangement from Z-3-hexenal to *E*-2-hexenal under both in vitro (Fig. 2) and in vivo conditions (Fig. 6). While previous work showed that the insect-induced re-arrangement of GLVs can alter the behavior of second and third trophic-level organisms[26,27], the present study furthermore indicates that the insect derived enzyme that we identified most likely also has other physiologically important functions that are essential for larval development and metamorphosis.

We observed developmental deficiencies in *M. sexta* *Hi-1* mutants fed on artificial diet that did not contain any plant GLVs (Fig. 7 and Supplementary Fig. 10). This indicates that the observed phenotypes in the *Hi-1* mutants were not caused by a lack of E2AL, and thus an inability to re-arrange Z3AL to E2AL, but that other, most likely intrinsic compounds of the insect must serve as a substrate for Hi-1, causing a growth effect in the mutants. At this point we can only speculate about the possible function of Hi-1 for the insect's own physiology. As the expression of Hi-1 is restricted to the salivary (labial) glands, the catalytic reaction of Hi-1 must happen either in the gland lumen, or on the plant while chewing, and subsequently in the gut lumen while digesting. Instead of the rather unusual function of a GMC oxidoreductase that we observed for Hi-1, i.e. the re-arrangement of the position of a double-bond from a *Z*-3- to an *E*-2-aldehyde, the main function of the enzyme might involve the simple catalysis of an oxidation-reduction reaction, often described for GMC oxidoreductases[42], in which a CH-OH group acts as a hydrogen or electron donor. Such bi-functional GMC oxidoreductase activity has also been described for the cholesterol oxidase, which catalyzes both oxidation and isomerization of cholesterol to 4-cholesten-3-one[43–45].

In any case, it is known that such deformations we identified at higher rates in the *Hi*-1 mutants, including less thoracic scales, unexpanded wings as well as proboscis abnormality (Fig. 7d), occur more frequently in malnourished insects[46–50]. Also, several substrates of GMC oxidoreductases are known to play essential roles in insect development such as glucose, choline and ecdysteroids[51–53], and future work using metabolomics will reveal whether Hi-1 is involved in any of the metabolic processes important for insect development.

By uncovering well-supported clustering with *M. sexta* Hi-1 (Fig. 8b, blue dashed circumferential line), phylogenetic analysis of the GMC oxidoreductase multi-gene family identified several candidate GMC homologs not only in *B. mori*, but also in *C. virescens*, *S. litura*, and *D. plexippus* that could possess hexenal isomerase activity. However, no clear one-to-one orthology was observed within this well-supported cluster as for instance *B. mori* BmHi did not cluster closely with *M. sexta* Hi-1 and Hi-like despite *B. mori* and *M. sexta* being the closest relatives of our species panel. The ability of *B. mori* BmHi to catalyze Hi activity and the inability of *M. sexta* Hi-like suggests that Hi activity might have an evolutionary history within Lepidoptera that is characterized by gain- and/or loss-of-function events. Although it remains unclear why and how *M. sexta* Hi-like lost Hi activity, pairwise alignment of the AlphaFold-predicted structures of MsHi-1 and MsHi-like suggests that the C-terminal extension of Hi-like might have caused a change in substrate binding affinity (Supplementary Fig. 16). However, these gain- and/or loss-of-function scenarios need to be approached with caution and require functional validation of a wider range of candidate Hi homologs before more solid conclusions can be drawn. Moreover, Hi activity could also have evolved outside this well-supported GMC cluster that contains *M. sexta* Hi-1. Indeed, a growing body of work shows that plants and animals can evolve the same enzymatic abilities by independent, convergent evolution within a single multi-gene family. Such convergent evolution happened for example in different clans of the cytochrome P450 multi-gene family possessing independent origins of keto-carotenoid and cyanogenic glycoside biosynthesis[54,55].

Previous work has shown that not only lepidopteran larvae[24,26,27], but also many plant species have the capacity to catalyze the conversion from *Z*-3-hexenal to *E*-2-hexenal[28,29,56–58]. Plant hexenal isomerases have been identified amongst others in bell pepper, potato, tea and cucumber plants[28,29,59]. However, the genes encoding for plant-derived hexenal isomerases belong to the family of cupin-like proteins and are thus part of a different multi-gene family than the one identified in this study. Clearly, these plant- and insect-derived enzymes that show identical function have evolved through convergent evolution, but it remains an open question whether there is divergent functionality between these two enzymes as well. The work on plant hexenal isomerases revealed that also other *Z*-3-alkenals can serve as substrate for this enzyme; early work on partially purified hexenal isomerase from cucumbers already showed that the enzyme affected the *Z*-3- to *E*-2-isomerization of several unsaturated *Z*-3-aldehydes with varying chain lengths[56]. Substrate promiscuity was also confirmed in recent studies in which recombinant protein of plant hexenal isomerases were used[28,29]. Similar substrate promiscuity was also observed for *M. sexta* Hi-1 which was not only able to convert *Z*-3-hexenal to *E*-2-hexenal, but also other Z-3-aldehydes of different chain lengths (Supplementary Fig. 5).

The subcellular localization of Hi-1 is predicted to be extracellular in contrast to the plant Hi which is predicted to be localized in the cytoplasm (Fig S15a, analyzed by DeepLoc-1.0[60]). Interestingly, the protein sequence of *M. sexta* Hi-1 does not contain a predicted signal peptide (Supplementary Fig. 15b, analyzed by SignalP-5.0[61]), indicating that its secretory pathway might be ER/Golgi-independent. The analysis by SecretomeP 2.0[62,63] further confirmed that Hi-1 secretion most likely occurs via the unconventional protein secretion pathway (Hi-1 score = 0.666, above the cut-off of 0.6). Several studies in insects have shown that large numbers of salivary (labial gland) proteins are devoid of any typical, predicted signal peptides, e.g. 70% of secreted saliva proteins in whitefly (*Bemisia tabaci*) and 27% of cocoon proteins (secreted from labial glands) of *B. mori* lack a predicted signal peptide[64,65]. Studies in *Drosophila* have shown that salivary gland proteins can be secreted through the apocrine mechanism, a non-vesicular process in which proteins are released into the gland lumen along with part of the cell[66,67]. This process can occur without the presence of a signal peptide[68].

One limitation of our study is that we were only able to keep one *Hi-1* mutant line to examine the developmental phenotypes. However, we have minimized the possibility of an off-target effect by back-crossing the *Hi-1* mutant twice to the wildtype strain before phenotyping[69,70]. Furthermore, we did not observe any off-target effect on the gene duplicate, *Hi-like*, even though 78% of gRNA1 (18 out of 23) and 87% of gRNA2 (20 out of 23) sequences were homologous to the last exon of *Hi-like*. In this study, we furthermore focused specifically on the developmental phenotypes of caterpillars that fed on artificial diet instead of on their host plant. While our study addresses the long-standing question about why *M. sexta* needs such maladaptive enzyme that causes the attraction of their own natural enemy, future work with plant-fed *hi*-mutants will reveal whether there are other synergistic or antagonistic effects on insect development that are caused by a plant-derived, and thus GLV-containing diet. Such a diet-dependent influence of Hi-1 on the insect's development might be possible, as previous results have shown that GLVs can indeed influence the feeding activity and survival of *M. sexta*[71,72].

Our findings offer further insights into why *Manduca* produces an enzyme that generates volatiles that can expose the insect to its predators, and why evolution has not eliminated this isomerase. While our previous research demonstrated that Hi-induced changes in the GLV-profile can act as a host-location cue for female adult moths[26], our current study shows that the hexenal isomerase likely plays a role in the insect's development as well. The occurrence of Hi-activity in several other lepidopteran species supports the hypothesis that this enzyme may have an either vital or non-vital but beneficial function(s) for the caterpillar that outweighs the costs of maintaining Hi-induced changes in host plant GLV-profile. If the enzyme serves a vital primary function for the caterpillar's physiology, as described here, it would prevent rapid selection against it.

The fact that herbivore- but also pathogen-derived elicitor compounds are generally present across species indicates that these have beneficial functions which soften the selection on them. For instance, flagellin and chitin act as pathogen-associated molecular patterns (PAMPs), but they also have essential functions for motion and

structural integrity in bacteria and fungi, respectively[73]. The long persistence of fatty-acid amino acid conjugate (FAC) elicitors in plant-herbivore interactions did not have an obvious explanation until Yoshinaga, et al.[74] reported that FACs are crucial for nitrogen assimilation and glutamine storage, and thus for the insect's survival. Thus, it is crucial to identify the full range of physiological properties of elicitors and modulators, as this knowledge is necessary for understanding why these molecules emerge, persist, disappear, and potentially reappear over time, and how they impact species interactions.

Future work will reveal whether the insect-derived hexenal isomerase has a similar substrate diversity to that of plants, whether a smaller subset of plant-derived substrates is used by insect hexenal isomerases, or whether the insect's own metabolites are major substrates instead. Addressing this question will help us to identify the main function of this enzyme for lepidopterans, thereby leading to a better understanding of why several lepidopteran species have retained an apparently poorly adapted enzyme in their genome[27].

## Methods

### Insects rearing

Tobacco hornworms (*Manduca sexta*) were obtained from the Max Planck Institute for Chemical Ecology (Jena, Germany). Colonies were maintained in the climate chamber under a 16 h/8 h (light/dark) cycle at 26 °C and 60% relative humidity. Collected eggs and hatched larvae were maintained in 250 mL plastic boxes (82 mm length x 108 mm width x 45 mm height) and fed *ad libitum* with an artificial diet. Individual pupae were kept in paper bags until emerging. Moths were maintained in insect cages and supplied with 20% sugar solution. Tobacco leaves (*Nicotiana tabacum*) were added into the cages to stimulate egg laying. Pupae of monarch butterflies (*Danaus plexippus*) were purchased from Costa Rica Entomological Supply and maintained in an incubator under a 16 h/8 h (light/dark) cycle at 22 °C and 70% relative humidity. Milkweed plants (*Asclepias incarnate*) were used for egg laying and larvae feeding. Cabbage white (*Pieris rapae*) were fed on *A. thaliana* (Col-0) and maintained in the climate chamber under a 16 h/8 h (light/dark) cycle at 26 °C and 60% relative humidity. Details of rearing conditions of other lepidopterans are described in the following articles: tobacco budworm (*Chloridea virescens*) in Lievers, et al.[75], tobacco cutworm (*Spodotera litura*) in Chang, et al.[76], and silkworm (*Bombyx mori*) in Lin, et al.[77].

### *M. sexta* artificial diet preparation

First, 100 g of agar was stirred well with 600 mL distilled water and pre-boiled 1600 mL water. The agar solution was boiled in a microwave until the agar was melted. While boiling the agar solution, another mixture was prepared, containing 1700 mL water, 300 g cornmeal, 50 g Wesson Salt Mixture, 75 g sugar, 10 g cholesterol, 315 g wheat germ, 160 g soy flour, 160 g casein, 25 g ascorbic acid, 12.5 g sorbic acid, and 6.7 g methylparaben, 20 g of vitamin mix and 20 mL of linseed oil. Finally, the melted agar solution was cooled down to below 60 °C, and both solutions were mixed. After thoroughly mixing, the solution was poured into plastic boxes with lids and exposed under UV light for 30 min before preserving in the cold room. The resources of compounds are listed in Supplementary Data 3.

### Plant rearing and collections

*Nicotiana attenuata*, *Nicotiana tabacum* and *Asclepias incarnate* were grown in the greenhouse with a day/night cycle of 16 h (26–28 °C)/8 h (22–24 °C) under supplemental light from Master Sun-T PIA Agro 400 or Master Sun-T PIA Plus 600-W sodium lights (Philips). Leaves of white mulberry tree (*Morus alba*) were collected at National Taiwan University (Taipei, Taiwan) or De Nieuwe Ooster cemetery (Amsterdam, Netherlands). *A. thaliana* (Col-0) were grown in the climate chamber under 11 h/13 h (light/dark) cycle at 22 °C and 70% relative humidity.

### Collection of insect's oral secretions (OS)

Larval OS was collected on ice using a small vacuum pump or glass capillary tubes as described in Roda, et al.[78]. Collected OS was centrifuged at 4 °C for 5 min at 13,000 g and the supernatant was transferred to a new tube, flushed with $N_2$ and snap frozen in liquid nitrogen. Samples were stored at −80 °C until use. Total protein concentration was determined by Bradford assay (Bio-Rad) following manufacturer's instructions, using BSA as standard.

### Partial purification of (3*Z*):(2*E*)-hexenal isomerase activity from *M. sexta* oral secretion (OS)

Gel Filtration was performed on a FPLC system (GE healthcare) equipped with a HiPrep 26/60 Sephacryl S-300 High Resolution column (GE healthcare), and protein elution was continuously monitored by UV absorption at 280 nm. To remove small particles, 2 mL of *M. sexta* OS was pushed through a 0.45 µm filter. One mL of the filtered solution was diluted 1:1 (v/v) with Tris buffer (100 mM NaSO₄, 40 mM Tris pH 9.0) and applied to the column which had been equilibrated with starting buffer (50 mM NaSO4, 20 mM Tris pH 9.0). The proteins were eluted isocratically at a constant flow of 1 mL/min. All fractions were tested for hexenal isomerase activity. The pooled active fractions (fraction 29–42) and two pools of non-active fractions (fractions 47–57; 69–70) were first concentrated with centrifugal filters (50 kDa cut-off, Ultracell Plus-70, Millipore) and proteins were subsequently precipitated with 80% acetone and re-suspended in MiliQ-water. The re-suspension volume for each sample was adjusted depending on the estimated amount of total protein per pooled fraction (between 20 and 150 µL). Samples were snap-frozen in liquid nitrogen and stored at −80 °C until further use. SDS-PAGE was done with Hoefer Mighty Small SE250 minigel equipment (Amersham Biosciences AB, Uppsala) using the Tris/Tricine buffer system and 10% acrylamide[79]. Silver staining was used to visualize proteins.

### Mass spectrometric analysis and identification of oral secretion proteins

The three slices of Ag-stained PAGE bands from the active fraction were treated with the in-gel digestion and extraction method as described by Shevchenko, et al.[80]. The collected eluates of each sample were dried and resuspended in 8 µl 50% ACN, 2% formic acid. The sample was then dried again in a speedvac and injected and analyzed using a volume of 25 µl 3% ACN, 0.1 % TFA. Peptide mixtures were analyzed by injection onto a reversed-phase capillary column (150 mm×75 um Pepmap C18, Thermo Scientific) at a flow rate of 300 nl/min by an Ultimate nano-LC-system (Dionex, Sunnyvale, CA). All solvents were of ULCMS grade (Biosolve) and peptides were separated by a multi-step gradient (solvent A 0.1% formic acid in water, Solvent B 0.1% formic acid in 95% acetonitrile and 4.9% water) starting at 5% B going to 15% B in 12 minutes, followed by a rise to 45% B in 23 minutes and to 100% B in 10 minutes (which is held for 5 minutes before returning to initial conditions). Eluting peptides were electrosprayed using a steel emitter (New Objective) with a capillary voltage of 1800V, cone voltage of 35 V, and an extractor voltage of 3 V as source settings with source temperature set at 120 °C into a Q-TOF mass spectrometer (Micromass) for subsequent mass spectral analysis. Peptides were analysed by data-dependent acquisition with a survey spectrum acquired from 350–1500 m/z and precursors rising above the selection threshold (Top 3) selected for fragmentation by collision with argon as a collision gas at a pressure of $4 \times 10{-}5$ bar as measured on the quadrupole pressure gauge and fragmentation spectra collected from 50-2500 m/z.

Resulting raw spectral files were processed using the proteinlynx plugin from Masslynx (Waters corporation) into peak list files (.pkl) using default settings and each was searched against three different databases together with a common contaminants database as compiled by the Max Planck Institute using a local license of MASCOT

software (Matrixscience release 2.7). The search parameters were as follows: allowance of one missed cleavage, enzyme is Trypsin/P, fixed modification of carbamidomethyl cysteine and variable modification of oxidation on methionine, an error tolerance of 0.3 Da for calculated peptides and their corresponding MS/MS spectra. The compiled databases used consisted of an EST database from midgut tissue[81] and RNA-Seq data from different adult and larval tissues, including salivary glands and larval mouth parts. The raw data for the *M. sexta* midgut tissue obtained by 454 sequencings can be accessed at NCBI Short Read Archive (SRA) with accession number SRX005619 and the dbEST database with accession numbers GR918023–GR923049. For the *M. sexta* adult tissue samples, raw data have been deposited at the EMBL-EBI SRA with study accession PRJEB2429. Previously unpublished Illumina short read (RNA-seq) data have been deposited to the EBI short read archive under study accession PRJEB62040 with the following sample accession numbers: ERS15422319 (adult antennae), ERS15422322 (larval salivary gland), ERS15422320 (larval fat body), ERS15422321 (larval mouthparts). All the peptide sequences of candidate proteins were again confirmed by NCBI blast. The names of gene annotation were according to JHU_Msex_v1.0[30], Supplementary Data 1 presents the list of candidate proteins.

## Cloning and construct design

The cDNA libraries of *M. sexta* were obtained from RNA-Seq databases and from the Msex_1.0 genome assembly[82]. The ORF of *Hi-1* was PCR amplified from cDNA of salivary (labial) glands of 4th instar larvae and cloned in the pJET1.2 vector (Thermo Fisher). Primers containing the attB1 and attB2 Gateway recombination sites were used to amplify ORF sequences. The resulting PCR products were recombined with the Gateway vector pDONR207 (Clontech) using BP Clonase II (Thermo Fisher). All constructs were verified by PCR and sequencing prior to LR-reaction with the destination vector. For the transient expression of *Hi-1* in *N. benthamiana* leaves and stable expression of *Hi-1* in Arabidopsis, the *Hi-1* cDNA were introduced into the destination vector pK2GW7 under control of the 35 S CaMV promoter (http://www.vib.be). For expression of Hi-1 recombinant protein in *E coli*, the cDNA of *Hi-1* was introduced into the destination vector pGEX-KG-GW[83]. LR reactions were performed with LR Clonase II (Thermo Fisher) and the resulting clones were confirmed by PCR and sequencing. All the above-mentioned plasmids were confirmed by sanger sequencing after cloning. For expression of the Hi-like, Xdh-like, Px-like 1, Px-like 2, Gld, and BmHi recombinant proteins in *E coli*, the cDNA sequences were synthesized and cloned into pGEX-4T-1 (Gene Universal), and the codon usages were adjusted to facilitate protein yield.

## Production of recombinant protein and purification

pGEX plasmids were transformed in *E. coli* BL21 star (DE3). A single colony of transformed *E. coli* was picked from LB plate and cultured in 10 mL LB medium shaking overnight at 37 °C. The overnight *E. coli* culture was then transferred to 1 L of 2 × YT medium (16 g tryptone, 10 g yeast extract, 5 g NaCl) and kept shaking at 37 °C. After *E. coli* grew to the optical density 0.4-0.5 (600 nm), IPTG (1 mM) was added for recombinant protein induction and shaking at 16 °C for 24 hours. The *E. coli* pellet was collected in a 50 mL falcon tube by repeated centrifugation, snap-frozen in liquid nitrogen and stored at −20 °C until use. 100 μg/mL of ampicillin was added to LB plates or medium. For purification of recombinant proteins, *E. coli* pellet was first resuspended in 10 mL of 1× PBS (pH 7.3) containing EDTA (1 mM), lysozyme (10 mg/mL) and proteinase inhibitor cocktails (50 mL per tablet). The supernatant was sonicated on ice (20 sec sonication and 10 sec rest for 6 times). Afterwards, 1% Triton X-100 and 2 uL of DNase I were added, and the tube was kept rotating for 30 min. The supernatant of *E. coli* lysate was collected by centrifugation (1600 g) and passed through a 0.45 μm filter. The GST-tagged candidate proteins were batch purified by affinity with GST Sepharose following the manufacturer

instructions. In brief, 10 mL lysate was first bound to 1 mL of GST sepharose overnight at 4 °C, washing sepharose with 1× PBS (pH 7.3) three times, and eluted with 50 mM Tris-HCl buffer (pH 8.0) with 10 mM GSH. All fractions were collected and run on SDS-PAGE to evaluate the yield and purification. Recombinant proteins were further visualized by western blot, and finally preserved in 50 mM Tris-HCl buffer (pH 8.0) with 10% glycerol at −80 °C until use. Densitometry was used to determine protein concentrations by comparing the relative intensity of bands between recombinant proteins and BSA standard with ImageJ (Supplementary Fig. 17).

## Western blot analysis

Protein samples (100 ng) were boiled in 4× loading buffer for 3 min, then loaded and run on an SDS-PAGE gel. Proteins were transferred to an Immobilon-E PVDF membrane (Millipore) by semi-dry blotting, and the membrane was washed three times with 1× PBST (0.05% Tween 20) for 15 min and blocked in 5% BSA for 1 h at room temperature before staining. The membrane was subsequently stained with GST-HRP conjugated antibody (1:2000) overnight at 4 °C in the cold room. After washing the membrane three times with 1× PBST (0.05% Tween 20) for 15 min, the membrane was treated with 1 mL of chemiluminescence solution (9 mL of $H_2O$, 1 mL of 1 M Tris-HCl pH 8.5, 22 uL of 90 mM p-coumaric acid, 50 uL of 200 mM luminol and 3 uL of 30% $H_2O_2$). Images were recorded using Odyssey® Fc Imaging System (LI-COR) and analyzed with Image Studio Lite (ver 5.2).

## Ectopic expression of *Hi-1* in *N. benthamiana* leaves by transient transformation

Agro-infiltrations were performed with 4-week-old *N. benthamiana* plants. *Agrobacterium tumefaciens* GV3101 (pMP90) cultures carrying the *Hi-1* constructs (pK2GW7: p35S-Hi-1 or a construct carrying only GFP (pK7WGF2.0)[84] were grown overnight from a single colony and diluted in infiltration buffer (¼ (v/v) LB medium, ¼ (v/v) sterile $H_2O$, ½ (v/v) 2x MS medium in 10 mM MES pH 5.6, 20 mM glucose, 10% (w/v) sucrose, and 200 μM acetosyringone) to an $OD_{600}$ of 0.6. To suppress gene silencing, each construct was co-infiltrated in a 1:1 ratio with an *A. tumefaciens* GV3101 (pMP90) strain that carried the pBIN61 vector to express the P19 suppressor[85]. Two leaves of each six plants were infiltrated with either the *Hi-1* construct or the GFP control construct. Another six plants were left untreated as an additional control treatment. Four days after infiltration, one leaf of each plant was harvested, snap-frozen in liquid nitrogen, and stored at −80 °C until further use.

## Ectopic expression of *Hi-1* in Arabidopsis thaliana by stable transformation

The construct, carrying the ORF of *Hi-1* under the control of the CaMV 35 S promoter, (pK2GW7: p35S-Hi-1) was transformed into *A. tumefaciens* strain GV3101. *A. thaliana* Col-0 was transformed using the floral dip method[86]. The seeds from primary *Arabidopsis* transformants were selected on 0.5 × MS medium containing 50 mg/L kanamycin and 20 mg/L Nystatin. T3 lines were used for characterization, and the transformation efficiencies were verified by RT-qPCR and by testing hexenal isomerase activity.

## Leaf sample preparation for Hexenal Isomerase Assay

Crude leaf extracts from *N. benthamiana* leaves and *A. thaliana* (Col-0) shoots were produced as follows: single leaves (or the whole shoot, in case of *Arabidopsis*) were ground on liquid nitrogen with a mortar and pestle and 1 mL of extraction buffer (20 mM MOPS pH 7.5, 0.2% Tween-20, 10 mM DTT) was added to the powder. The solution was gently shaken for 20 min and subsequently centrifuged at 13,000 g at 4 °C for 30 min. The supernatant was transferred to a new tube and snap-frozen in liquid nitrogen and stored at −80 °C until further use. To determine hexenal isomerase activity, we used either 40 μL (*N.*

*benthamiana*) or 200 μL (*Arabidopsis*) of leaf extract per analysis. Protein concentration was determined by Bradford assay.

## Z-3-nonenal and Z-3-octenal synthesis

The synthesis of *Z*-3-nonenal and *Z*-3-octenal were done according to the literature with some modifications[87]. Dess-Martin periodane (1.27 g, 1.5 Eq, 3.00 mmol) and 20 mL dichloromethane (DCM) were added in a flame dried round-bottom flask and stirred. *Z*-3-octenol (256 mg, 0.302 mL, 1 Eq, 2.00 mmol) or *Z*-3-nonenol (284 mg, 0.33 mL, 1 Eq, 2.00 mmol) was added at once and the reaction was followed by thin layer chromatography (TLC). After one hour, all starting material was gone and TLC indicated 1 spot. 0.5 M sodium thiosulphate in saturated sodium bicarbonate in water (20 ml) was added and stirred for 5 minutes and subsequently extracted with DCM (3 times 20 ml). The combined organic fractions were dried over anhydrous magnesium sulfate, filtered and evaporated. The dried crude was dissolved in pentane and filtered over cotton and evaporated to obtain *Z*-3-octenal (221 mg, 1.75 mmol, 88 %) or *Z*-3-nonenal (243 mg, 1.73 mmol, 87 %) as a colorless oil. NMR and mass spectrums were taken to confirm synthesized substrates. Chemical structures and NMR data are provided in the Supplementary Fig. 18.

## Isomerase activity assays

Enzyme activity was determined by SPME-GC-ToF-MS or SPME-GC-qToF-MS. 200 μL of solution, either containing diluted oral secretion, crude leaf extract, purified fractions, or purified recombinant proteins in a 20 mM Tris-HCl buffer (pH 8.5), 20 μg of BSA and varying amounts of volatiles were transferred to a 1.5 mL GC vial equipped with a 200 μL insert. A volume of 200 μL was chosen to minimize the headspace and thus also the chance of volatilization. The GC vial was closed and gently shaken for 2 min (recombinant protein and diluted OS) or 5 min (collected fractions from partial purification and crude leaf extract). Subsequently, the mixture was transferred to a 20 mL SPME vial which was immediately closed with a Teflon lined crimp cap.

## Analysis of volatiles

Volatiles were initially analyzed by GC-ToF-MS (samples from the purification), until we obtained a new more sensitive GC-qTof-MS. In all cases volatiles were sampled with a Solid Phase Micro Extraction fiber (SPME; Carboxem/PDMS) for 10 min at 35 °C.

## Analysis of volatiles by GC-ToF-MS

After sampling for 10 min, the fiber was desorbed for 1 min in an Optic injector port (ATAS GL Int. Zoeterwoude, NL) which was constantly kept at 250 °C. Compounds were separated on a DB-5 column (10 m x 180 μm, 0.18 μm film thickness; Hewlett Packard) in a 6890 N gas chromatograph (Agilent, Amstelveen, NL) with a temperature program set to 40 °C for 1.5 min, increasing to 250 °C at 30 °C per min and 250 °C for an additional 2.5 min. Helium was used as carrier gas, with the transfer column flow set to 3 mL per min for 2 min, and to 1.5 mL per min thereafter. Mass spectra were generated by electron ionization with 70 eV electrons at 200 °C and collected with a Time-of-Flight MS (Leco, Pegasus III, St. Joseph, MI, USA), with an acquisition rate of 20 scans per second.

## Analysis of volatiles by GC-qToF-MS

After sampling for 10 min the fiber was desorbed for 1 min in the injection port which was constantly kept at 250 °C. Compounds were separated on HP-5ms column (30 m x 250 μm, 0.25 μm film thickness; Agilent) in an Agilent 7890 A gas chromatograph with a temperature program set to 40 °C for 5 min, increasing to 140 °C at a rate of 5 °C per min, followed by increasing temperature to 250 °C at a rate of 15 °C per min and an additional 5 min at 250 °C. Helium was used as the carrier gas with the transfer column flow

set to 3 mL per minute and a flow rate of 1 mL per min thereafter. Mass spectra were generated by an Agilent 7200 accurate-mass quadrupole time-of-flight mass spectrometer, operating in electron ionization mode (70 eV) at 230 °C and collected with an acquisition rate of 5 scans per second. Volatiles were identified and quantified using standard volatiles listed in Supplementary Data 3. To determine the conversion from *Z*-3-hexenal to *E*-2-hexenal we first calculated the sum of aldehydes (*Z*-3-hexenal + *E*-2-hexenal) measured by SPME, taking the response factors of each compound into account. We subsequently calculated the percentage of *E*-2-hexenal and subtracted the non-enzymatic conversion from this value.

## Kinetics

In order to determine the kinetic parameters of MsHi-1, purified GST-tagged protein was used (see above for details of the assay). Purified GST-tagged protein (-90 kDa; 37.5 ng) was diluted in reaction buffer (20 mM Tris-HCL, pH 8.5, 2 μL BSA (10 mg/mL), 50 mM FAD, 100 mM NADH) to a final volume of 200 μL. The substrate *Z*-3-hexenal was added to the reaction mixture in a concentration range of 500–8000 μM and the mixture was incubated for 2 min at RT while gently shaking. To stay within the linear range of the instrument (GC-qToF-MS) an aliquot (6.3–100 μL) of the reaction mixture containing an equivalent of 250 μM *Z*-3-hexenal was transferred to a 20 mL SPME vial, immediately closed and volatiles were absorbed onto the fiber for 10 min. The fiber was subsequently injected into the GC and volatiles were analyzed as described above. The conversion from *Z*-3-hexenal to *E*-2-hexenal was calculated as described above. Absolute values of *E*-2-hexenal were then re-calculated by taking the initial concentration of *Z*-3-hexenal used for the assay into account. $K_m$ and $K_{cat}$ values were determined from Lineweaver-Burk Plots ($n = 3$ biologically independent samples).

## Gene expression analysis

For the RNA extraction of *Arabidopsis* stably expressing *Hi-1*, we harvested material from 5-week-old *Arabidopsis* plants (whole shoot). For RNA extraction from different larval tissues, the fourth instar *M. sexta* were first anesthetized on ice and dissected in cold 1× PBS (pH 7.3). Tissues from three individuals were pooled, representing one biological replicate. For the RNA extraction from different developmental stages of *M. sexta*, 20 eggs or 10 individuals of neonates, 1st instar, 2nd instar, or 5 individuals of 3rd instar, 4th instar, 5th instar, pupae or 5 adults, respectively, were pooled per RNA sample. The caterpillars were reared on an artificial diet. Both sexes (three males plus two females or vice versa) were pooled for each pupa or adult RNA sample. All samples were ground in liquid nitrogen with a pestle and mortar, kept frozen after collection and were stored at -80 °C until proceeding with RNA extraction.

Total RNA was extracted from ground materials using the TRIzol/chloroform method. Due to RNA impurities contained in *M. sexta* pupal samples, after phase separation by TRIzol/chloroform treatment, the RNA supernatant was further purified using RNA columns (QIAGEN) according to the manufacturer's protocol. All of the purified RNA samples were DNase treated to avoid genomic DNA contamination. cDNA was synthesized from 1 μg (plants) or 1.5 μg (insects) of total RNA using the RevertAid transcriptase. Quantitative real-time PCR (ABI 7500 Real-Time PCR System; Applied Biosystems) was done by using the HOT FIREPol® Eva-Green® qPCR Mix Plus. The *Hi-1* expression level in *M. sexta* was calculated using the $2^{-\Delta\Delta Ct}$ method with the ct values of *Hi-1* normalized to a reference gene, *ubiquitin*. The *Hi-1* expression level in transgenic *Arabidopsis* was calculated using the $2^{-\Delta Ct}$ method with the ct values of *Hi-1* normalized to a reference gene, *sand*. All the resources of chemicals and primer sequences are listed in Supplementary Data 3.

### CRISPR/Cas9 mutagenesis in *M. sexta*

The details of embryo preparation, microinjection and genotyping were described by Fandino, et al. [88]. The procedure was done at the Department of Evolutionary Neuroethology (Max Planck Institute for Chemical Ecology, Jena, Germany). Selection of guide RNA (gRNA) target sites and off-targeting prediction were analyzed using CHOP-CHOP (http://chopchop.cbu.uib.no)[89–91]. *Streptococcus pyogenes* Cas9 with nuclear localization sequence (NLS) and two gRNAs of *Hi-1* (gRNA1: 5′-CCTCGCAGAATACGCCTCCCATC-3′; gRNA2: 5′-AGAATG-GAGGCAAAGCGCTGCGG-3′) were ordered from IDT Inc. The molar ratio was 1:8 for 4 µM of Cas9 (640 ng/uL) to 32 uM of gRNA (365 ng/uL) in the injection solution. For genotyping, DNA was extracted by cutting the larval horn, and MyTaq Extract-PCR Kit was used following the manufacturer's instructions. For sanger sequencing, the Phusion® High-Fidelity DNA Polymerase with proofreading activity was used. All the resources of chemicals are listed in Supplementary Data 3.

### Plant mechanical wounding and oral secretion treatment

A 24 mm diameter of leaf disc was punched out from leaf lamina at S4 leaf position[92] of five weeks old *N. attenuata plants*. The leaf disc was mechanically wounded with a fabric pattern wheel and 10 µL OS (10 µg total protein) or water was applied to the wounds and gently dispersed across the leaf surface. After 20 secs, the leaf disc was transferred to an SPME vial, and volatiles were immediately collected with a Solid Phase Micro Extraction fiber (SPME; Carboxem/PDMS) for 10 minutes at 35 °C and measured by SPME-GC-qToF-MS analysis as described above.

### Phenotyping *Hi-1* mutants

To minimize genetic background effects[70,93], the *Hi-1* mutant strains (*Hi-1⁻*) were backcrossed to wildtype strains (*Hi-1⁺*) and homozygous *Hi-1* mutants were obtained before phenotyping. For the comparison of developmental times, the duration from neonate (first day of hatching) to adult (first day of emerging) was recorded. For the comparison of pupal and adult weights, the pupae from 6 days after wandering stages and the adults from first day of emerging were weighed individually on an electronic scale (SE622, VWR). The morphological deformations including proboscis misfolding, loss of thoracic scales, and unexpanded wings from newly emerged adults were recorded. The phenotypes were continuously recorded for three consecutive generations at 26 °C, and one generation at 24 °C.

### GMC identification and phylogenetic analysis

To identify GMC genes in the annotated genome assemblies of *M. sexta*, *P. rapae*, *D. plexippus*, *C. virescens*, and *S. litura*, profile-HMM searches were performed against the predicted proteomes using the GMC_oxred_N profile (PF00732.19) and the default inclusion threshold (version 3.2.1)[30,94–98]. Reference GMC protein sequences were selected and retrieved from previous work[31,34,99,100]. The analyzes did not include the reference *B. mori* protein sequences of BGIBMGA012996, BGIBMGA013006, and BmEO due to unavailable public annotation files. The final set of 381 GMC protein sequences were aligned using MAFFT with the E-INS-I strategy (combination of --genafpair and --maxiterate 1000) (version 7.310) Supplementary Data 2[101]. The LG + R9 protein model was selected using ModelFinder based on the Bayesian Information Criterion[102]. Maximum-likelihood tree reconstruction was performed with IQ-TREE version 2.1.2 (random seed number was set at 12345)[103]. Ultrafast bootstrapping was implemented with 3000 replicates, nearest neighbor interchange (-bnni), and a minimum correlation coefficient of 0.99 (-bcor 0.99)[104]. The phylogenetic tree was rooted using the three fungal GOX genes. A subset of 61 GMCβ protein sequences was selected based on the phylogenetic reconstruction and aligned as described above. The number of

ultrafast bootstrap replicates was increased to 4500 to generate the second phylogenetic tree. The *M. sexta* GMCβ protein sequence of XM_030178129.1 was used to root the second tree.

### Statistical analysis

Statistical analyzes were performed using GraphPad Prism 9. Shapiro-Wilk test was applied for analyzing the normal distribution of values. For comparisons between multiple groups, one-way ANOVAs followed by Tukey's multiple comparison tests were used. Different letters in the graphs above each group indicate significant differences between groups ($p < 0.05$). For the comparisons of phenotypes (developmental times, body weight, and adult deformation) between wildtype and *Hi-1* mutants of each sex, two-tailed unpaired t-tests were performed. For the analysis of hexenal isomerase activity in the OS from different lepidopterans or the crude protein extract from their host plants (Supplementary Fig. 12), Mann–Whitney U test was used to compare the proportion of *E*-2-hexenal between samples and buffer control. Error bars are given as standard error of the mean (SEM). *$p < 0.05$, **$p < 0.01$, ***$p < 0.001$, n.s., not significant.

### Reporting summary

Further information on research design is available in the Nature Portfolio Reporting Summary linked to this article.

## Data availability

Source data are provided with this paper. Mass spectrometric analysis of oral secretion protein is available on MassIVE (Center for Computational Mass Spectrometry, UC San Diego) with accession number MSV000092011. All other data that support the findings of this study are available from the corresponding author upon request. Transgenic Arabidopsis seeds can be made available upon request. Source data are provided with this paper.

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

## Acknowledgements

We thank Dr. Erik Poelman (Wageningen University) who kindly provided us with cabbage white (*Pieris rapae*) eggs and Prof. Ian T. Baldwin who kindly provided us with large amounts of *M. sexta* OS. We appreciate the help of our colleagues at University of Amsterdam, Dr. Marc Galland for the discussion of statistical analysis, Dr. Gertjan Kramer for the discussion of LC/MS analysis, Ludek Tikovsky and Harold Lemereis for taking good care of all plants in the glasshouse, and Prof Jan van Maarseveen and Nick Westerveld for synthesizing the *Z*-3-alkenals. This work was supported by the European Research Council (ERC) under the European Union's Horizon 2020 research and innovation programme (grant agreement n° 805074) (S. A., Y. H. L. and J. S.), the EU grant FP7-PEOPLE-2011-IEF-302388 (S. A.) and NWO-ALW grant 863.14.011 (S. A.). N. W. was supported by a BOF post-doctoral fellowship (Ghent University, 01P03420).

## Author contributions

Y.H.L. participated in the design of the study, analyzed data, performed experiments, and drafted the manuscript. J.S. performed experiments, analyzed data, drafted the manuscript, and illustrated figures. N.W. analyzed the phylogeny and helped to draft the manuscript. R.A.F. performed the CRISPR/Cas9 mutagenesis. H.D. and C.D.K. performed LC/MS proteomic analysis. H.V. provided transcriptomic databases and helped to draft the manuscript. Y.L.W. collected the oral secretion samples. E.G.W. supervised the experiments related to CRISPR/Cas9 mutagenesis. M.H. and R.S. supervised experiments and helped to draft the manuscript. S.A. conceived the study, supervised the project, performed experiments, and drafted the manuscript. All authors read and approved the final manuscript.

## Competing interests

The authors declare no competing interests.
