## [Peer review file · Nature Communications]

REVIEWER COMMENTS

Reviewer #1 (Remarks to the Author):

In 2010, one of the authors has shown that a component in the oral secretion of *Manduca sexta* attracts its natural enemies more strongly by converting 3Z-hexenal released from the damaged part of the tobacco leaf into 2E-hexenal. In this study, the authors have identified MsHi-1 as the component responsible for this isomerization. The recombinant enzyme expressed in *E. coli* showed isomerization activity, the gene was expressed specifically in the salivary gland, and the activity in the OS was lost when the gene was disrupted by genome editing of *M. sexta*, thus confirming that Hi-1 is the isomerase in the *M. sexta* OS.

Elaborate experiments have demonstrated that Hi-1 is an isomerase in *M. sexta* OS. However, evidence that Hi-1 contributes to the ecological significance of the isomerase in nature published in *Science* in 2010 has not been provided. This story will be completed by demonstrating the involvement of Hi-1 in the plant-herbivore-predator tritrophic system by using the genome-edited *M. sexta* in nature (or at least in a nature-like environment). In examining the effect of Hi-1 on the composition of green leaf volatiles, it is essential to use the Hi-1 deficient *M. sexta* larvae to feed on tobacco leaves and compare the GLV composition produced with that infested by the wild-type strains. Examining the attractiveness of predators, e.g., *Geocoris* spp. to the volatiles formed from tobacco leaves infested by either the Hi-1 deficient strains or the wild-type strains will be able to be performed.

If Hi-1 is secreted into the OS, there must be a secretory signal in the gene. Please provide information on a signal sequence.

It is not appropriate to evaluate the purity of the recombinant enzymes with Western blotting. Total protein profiles should be shown with CBB or silver staining in Fig. 2.

Fig. 2B shows that the enzyme yielded about 10% of E2-hexenal at 1 μ g. However, the isomerization resulted in more than 50% E2-hexenal from the chromatograms shown in Fig. 2C. Why was this big difference in isomerization efficiency observed? The reason for this difference in efficiency needs to be shown.

Since Z3-hexenal can be isomerized to E2-hexenal non-enzymatically by a base, it seems to be essential to show the kinetic parameters, such as K_m or k_{cat} values, as data supporting that Hi-1 is an enzyme catalyzing the isomerization of Z3-hexenal.

The authors claim that the genes shown with the dashed line in Fig. 8A and the blue star in Fig. 8B as GMCs with Hi activity. However, it should be noticed that MsHi-like has no Hi activity even though MsHi-like locates next to MsHi-1 in the same sub-clade, and therefore, the basis for this claim is not valid. There is no evidence that the other six Ms genes found in the clade have Hi activity, for example. In order to make such a claim, it is necessary to clarify whether the products of at least several genes noted here have Hi activity.

In the Discussion section, the authors explain that the discrepancy between the results shown in this study and those of Jones et al. 2021a is due to the different instars of insects used in the assay, but this is not a sufficient explanation because the insects used in Jones et al. 2021a contained OS from third to fifth instars. It is recommended to carry out the isomerase assay with silkworms of various instars. The claim that the GMC homologs of the insects used in this study other than silkworms found in the same clade as MsHi-1 in the phylogenetic tree is also unacceptable because the activity of gene products other than MsHi-1 in the GMC homologs in this clade has not been verified. As mentioned above, it is essential to show clear evidence about the correlation between the location in the clade and Hi activity. To accomplish this, investigation of enzyme activities of recombinant proteins encoded by the genes found in the same clade should be carried out. Also, the reason why MsHi-like showed no

Hi activity should be provided.

I don't think the authors are providing any significant findings through the comparison of *M. sexta* Hi with plant His, because properties of MsHi-1 have not been fully investigated. For example, the substrate specificity of MsHi-1 should be examined and the results should be compared with those of plant His.

The size-exclusion membrane ultrafiltration showed the enzyme should be bigger than 50 kDa, but is it possible that the enzyme consists of subunit structure? If, for example, the enzyme would be a dimer, then, the monomeric unit should be 25-50 kDa. In the sense, the claim that the monomeric structure of the enzyme should be bigger than 50 kDa is inappropriate, at least, before identification of the gene. It would be helpful to estimate molecular weight of the enzyme with the chromatogram obtained with Sephacryl S-300.

Reviewer #2 (Remarks to the Author):

Key results

In the paper 'A salivary GMC oxidoreductase of *Manduca sexta* re-arranges the green leaf volatile profile of its host plant' a (3Z):(2E)-hexenal isomerase (Hi-1) was identified from the salivary glands of *M. sexta*. This protein not only alters the GLV profiles from host plants, but also is involved in interesting physiological ramifications for the insect during development. The authors also found Hi-1 homologs in other lepidopterans that could catalyze similar reactions.

The purification and identification of the previously described (3Z):(2E)-hexenal isomerase as belonging to the GMC β subfamily represents a significant addition to the field of insect oral secretions and plant GLV emission. The further discovery that Hi-1 knockout mutant adults experienced higher amounts of morphological deformations such as proboscis misfolding, loss of thoracic hair, and non-expanded wings, demonstrates that Hi-1 has an essential role in insect physiological development and the observed function in catalyzing GLV isomers is likely an indirect plant response to this protein.

Data Interpretation

Overall, the validity and robustness of the data interpretations and conclusions are strong and do not over-reach. The reference to the 'change in volatile signal is bittersweet for the insect as it can be used by their natural enemies as a prey location cue' and the referral to this insect-derived enzyme as a 'double-edged sword' (lines 24-25 and 86, respectively) could benefit from further discussion. Particularly, in light of the findings presented here that concern the role of this protein in insect physiological development, the paper would benefit from discussion of the placement of this protein in the context of the plant-insect 'arms race'.

Context of Results

The results of this study have been provided with sufficient context and consideration of previous work, as reflected in the introduction, discussion, and references.

The focus of this study was to identify the protein involved in the previously reported shift in the emission of GLV isomers, not to perfectly replicate a natural 'real world' scenario with herbivory on whole plants. However, normalizing Hi-1 activity by protein content (lines 288-299), does not account for physiologically or ecologically relevant protein concentrations present in caterpillar saliva or those necessary to isomerize (3Z):(2E)-hexenal in whole plants. Hi-1 activity and the concentration of this protein likely vary among caterpillar species. The determination and use of biologically relevant concentrations of Hi-1 from *B. mori* (and other caterpillar species) would clarify any differences in results between this study and those found in the literature.

Methods

This is an extensive study which utilized an impressive suite of techniques and methods to identify Hi-1 and address the role of this protein in isomerizing GLVs from host plants and in insect development, through the use of CRISPR-Cas9, cloning, phylogenetic analysis, VOC collection and analysis, and expression of Hi-1 in two host plants. The data and methods are presented in clear detail.

Considering the physiological defects in the CRISPR-Cas9 insect mutants (proboscis misfolding, loss of thoracic hair, and non-expanded wings) likely have a negative effect on insect reproduction and survival, it is impressive that these mutant lines could be maintained. Could further clarification be included to account for the maintenance of the CRISPR-Cas9 mutants lines for three generations?

Since GLVs have been successfully collected on adsorption polymers (such as Super Q or Hayesep Q) with dynamic push-pull techniques from both homogenized leaf tissue and whole plants (see: Engelberth J, Engelberth M. Variability in the capacity to produce damage-induced aldehyde green leaf volatiles among different plant species provides novel insights into biosynthetic diversity. *Plants*. 2020 9(2):213. and Allmann S, Baldwin IT. Insects betray themselves in nature to predators by rapid isomerization of green leaf volatiles. *Science*. 2010, ;329(5995):1075-8. among others), it would be beneficial to provide the rationale for using SPME to collect GLVs since VOC collection methodologies have different strengths and limitations. Furthermore, analysis of GLVs by GC-ToF-MS and GC-qToF-MS is less common in the literature than analysis by GC-FID/GC-MS. Why was this analytical method selected for volatile analysis?

Suggested improvements

The described leaf disc assay (lines 602-608) undoubtedly allows for highly controlled volatile collection from treated plant tissue. However, this extensive study would greatly benefit from testing the GLV emission from whole plants treated Hi-1 and OS from the various caterpillar species. Such an addition, as well as the use of biologically relevant Hi-1 concentrations from *M. sexta* saliva would provide a pertinent aspect to this study.

Line 156, 158 and elsewhere: specify which of the two types of caterpillar salivary glands were dissected, mandibular or labial.

REVIEWER COMMENTS

Reviewer #1 (Remarks to the Author):

In 2010, one of the authors has shown that a component in the oral secretion of *Manduca sexta* attracts its natural enemies more strongly by converting 3Z-hexenal released from the damaged part of the tobacco leaf into 2E-hexenal. In this study, the authors have identified MsHi-1 as the component responsible for this isomerization. The recombinant enzyme expressed in *E. coli* showed isomerization activity, the gene was expressed specifically in the salivary gland, and the activity in the OS was lost when the gene was disrupted by genome editing of *M. sexta*, thus confirming that Hi-1 is the isomerase in the *M. sexta* OS.

Elaborate experiments have demonstrated that Hi-1 is an isomerase in *M. sexta* OS. However, evidence that Hi-1 contributes to the ecological significance of the isomerase in nature published in *Science* in 2010 has not been provided. This story will be completed by demonstrating the involvement of Hi-1 in the plant-herbivore-predator tritrophic system by using the genome-edited *M. sexta* in nature (or at least in a nature-like environment). In examining the effect of Hi-1 on the composition of green leaf volatiles, it is essential to use the Hi-1 deficient *M. sexta* larvae to feed on tobacco leaves and compare the GLV composition produced with that infested by the wild-type strains. Examining the attractiveness of predators, e.g., *Geocoris* spp. to the volatiles formed from tobacco leaves infested by either the Hi-1 deficient strains or the wild-type strains will be able to be performed.

ANS: Thank you for your insight, we appreciate your comment. We believe our work provides a novel and important framework for establishing biological necessity of ecologically relevant candidate genes in the herbivore-prey relationship. Therefore, the main goals of this study are (i) to identify the enzyme in the oral secretions of *M. sexta* that converts the GLV Z3AL to E2AL and (ii) to understand its role in vivo relative to the full range of salivary enzymes. These results are a first step towards understanding why *M. sexta*, and possibly other lepidopterans, retain an apparently maladaptive enzyme in its gene pool.

We completely understand the request for ecological validation. However, limitations in current national and international policies for field experimentation using unclassified GMOs severely limits our current abilities to perform such experiments in natural settings. Alternatively, as the editor suggests, we could mimic nature-like conditions in the lab. Unfortunately, rearing of the *Geocoris* predator is challenging and we do not have the means and manpower to do this. Finally, the mutant colony was extremely difficult to propagate due to the physiological defects. This severely constrains the possibility to perform solid herbivore-predator studies that typically require a high level of replication.

If Hi-1 is secreted into the OS, there must be a secretory signal in the gene. Please provide information on a signal sequence.

ANS: We have investigated the presence of a signal peptide in the Hi-1 protein. Results from our analysis using SignalP-5.0 show that there is no classical secretory signal peptide in Hi-1 (Fig S15B). This suggests that Hi-1 secretion occurs through an ER/Golgi-independent

mechanism. Our analysis using SecretomeP 2.0, which predicts proteins secreted via non-classical pathways, confirmed that Hi-1 secretion most likely occurs through the unconventional protein secretion (UPS) pathway, with a score of 0.666 (above the cut-off of 0.6). Several studies in insect have shown that large numbers of salivary (labial gland) proteins are devoid of any typical, predicted signal peptides, e.g. 70% of secreted saliva proteins in whitefly (*Bemisia tabaci*) and 27% of cocoon protein (secreted from labial glands) of *B. mori* lack predicted signal peptides (Huang et al., 2020; Zhang et al., 2015). Studies in *Drosophila* have shown that salivary gland proteins can be secreted via the apocrine mechanism, a non-vesicular process in which proteins are released along with a portion of the cell into the gland lumen. (Farkaš et al. 2014; Farkaš 2015). This process can occur without the presence of a signal peptide (Aumüller et al. 1999). These findings have been added to the manuscript and are discussed in lines 327-338, which are highlighted in yellow.

It is not appropriate to evaluate the purity of the recombinant enzymes with Western blotting. Total protein profiles should be shown with CBB or silver staining in Fig. 2.

ANS: We understand the importance of evaluating the purity of recombinant enzyme extracts through total protein profiles and agree with the recommendation to include this information. While western blotting was already shown in the previous version, we now also provide the results of the SDS-PAGE with CBB staining. The SDS-PAGE gel was used to determine the concentration of recombinant protein through densitometry analysis, as described at line 512-513. We have now added the SDS-PAGE gel to the supplementary materials (Fig. S17) and included a reference to it in the Materials and Methods section at line 513 (highlighted in yellow).

Fig. 2B shows that the enzyme yielded about 10% of E2-hexenal at 1 µg. However, the isomerization resulted in more than 50% E2-hexenal from the chromatograms shown in Fig. 2C. Why was this big difference in isomerization efficiency observed? The reason for this difference in efficiency needs to be shown.

ANS: Thank you for the comment. We are aware of the difference in isomerization efficiency between Fig 2B and Fig 2C and would like to clarify the reason for this difference. The data in Fig 2B have been normalized using the corresponding response factor for each compound. In chromatography, a response factor is defined as the ratio between the concentration of a compound being analysed and the response of the detector to that compound. Such a normalization step is necessary to be able to calculate the total amount of aldehydes (and thus the percentage of E2AL) as the two measured compounds, Z3AL and E2AL have very different response factors when measured by GC-qToF; in other words: when we measure the same amount of either Z3AL or E2AL this will lead to a much greater peak area for E2AL than for Z3AL. This explains why the conversion to E2AL seems greater in Fig 2C (which represents unprocessed data). Here data are represented as relative ion counts (i.e. the compound that gives the highest ion count is set to 100% and all other compounds relative to it). The differences in the detector response have been accounted for in figure 2B (and thus not in figure 2C) as the data are normalized by using response factors.

We understand that the different visualization methods in the figures 2B and 2C can cause confusion and we have added clarification in the figure legend (line 736-738) to explain the normalization process used in Fig 2B and the raw data presentation in Fig 2C. We have also

added extra information on how we determined the percentage of *E*-2-hexenal in the Material and methods section in lines 602-605 (highlighted in yellow).

Since Z3-hexenal can be isomerized to E2-hexenal non-enzymatically by a base, it seems to be essential to show the kinetic parameters, such as K_m or k_{cat} values, as data supporting that Hi-1 is an enzyme catalyzing the isomerization of Z3-hexenal.

ANS: We agree with the reviewer's comment and we have now included the kinetic parameters of Hi-1 in Table S2. The results are now described at line 123-125 as well as in the material and method section at line 607-619 highlighted in yellow.

The authors claim that the genes shown with the dashed line in Fig. 8A and the blue star in Fig. 8B as GMCs with Hi activity. However, it should be noticed that MsHi-like has no Hi activity even though MsHi-like locates next to MsHi-1 in the same sub-clade, and therefore, the basis for this claim is not valid. There is no evidence that the other six Ms genes found in the clade have Hi activity, for example. In order to make such a claim, it is necessary to clarify whether the products of at least several genes noted here have Hi activity.

ANS: We believe an unfortunate misunderstanding is at the root of this comment. First, within the *M. sexta* species, since the *Hi-1* mutation abolished Hi activity entirely (Fig. 6), we are convinced that Hi-1 is the sole *M. sexta* enzyme contributing to Hi-activity in the oral secretion of *M. sexta* larvae. Second, we do not make the claim that all the homologs within the blue star clade have Hi-activity. We are fully aware that this clade includes genes, such as the MsHi-1 paralog, MsHi-like, that have clearly no Hi activity. Based on our results, we wanted to identify a set of **candidate** genes that could catalyse Hi reactions in other lepidopterans. Indeed, this was outlined throughout the manuscript, including the introduction section (e.g. "Together with OS activity assays for this lepidopteran panel, we identified candidate *Hi-1* homologs from other lepidopteran species that could catalyze similar GLV conversions." as well as the results section (e.g. "Based on these observations, we propose to narrow down candidate GMC genes that code for hexenal isomerase activity to a more internal node that contained 53 lepidopteran GMCs (indicated by a blue dashed circumferential line in Figure 8A and blue star in Figure 8B)"). In fact, to prove the validity of our candidate selection criteria, we now functionally expressed a *Bombyx mori* Hi candidate, *BGIBMGA000158* (Figure 8B). Recombinant protein of this Hi-candidate was able to convert Z3AL to E2AL (Figure S14). Finally, the observation that MsHi-like does not catalyze a Hi reaction is one of the reasons we indicated in our discussion at line 301-302, that "Hi activity (and similar enzymatic abilities) might have an evolutionary history within Lepidoptera that is characterized by gain- and loss-of-function events".

We believe our title of Figure 8 might have contributed to this misunderstanding. To avoid further confusion, we have rephrased the title as "Hexenal isomerase activity in oral secretions is pervasive in Lepidoptera and is associated with a specific cluster within the GMC β subfamily".

In the Discussion section, the authors explain that the discrepancy between the results shown in this study and those of Jones et al. 2021a is due to the different instars of insects used in the assay, but this is not a sufficient explanation because the insects used in Jones et al. 2021a contained OS from third to fifth instars. It is recommended to carry out the isomerase assay with silkworms of various instars. The claim that the GMC homologs of the insects used in this

study other than silkworms found in the same clade as MsHi-1 in the phylogenetic tree is also unacceptable because the activity of gene products other than MsHi-1 in the GMC homologs in this clade has not been verified. As mentioned above, it is essential to show clear evidence about the correlation between the location in the clade and Hi activity. To accomplish this, investigation of enzyme activities of recombinant proteins encoded by the genes found in the same clade should be carried out. Also, the reason why MsHi-like showed no Hi activity should be provided.

ANS: Thank you for this suggestion. As outlined in our above response, we do not state that (all) the GMC genes of the blue star clade encode for Hi enzymes. Instead, we used a combined phylogenetic/oral secretion assays approach to arrive at a list of **candidate** Hi genes.

To fully address the apparent discrepancy between our study and Jones et al., 2021a, we have now produced recombinant protein from one of the putative *Bombyx mori* Hi genes, *BGIBMGA000158*. Because of its tissue-specific expression in the labial glands of 4th-5th instar larvae, this candidate was already suggested in our discussion as a very likely Hi candidate (Fig. S13). Our in vitro experiment with recombinant BGIBMGA000158 confirms that this candidate has indeed Hi activity, as E2AL levels were increased when recombinant protein was incubated with Z3AL as substrate (Fig. S14). Although we cannot fully explain the difference in results reported between the two studies, our data clearly shows that *B. mori* possess an enzyme that is able to convert Z3AL to E2AL in vivo (Fig. 8C) as well as in vitro (Fig. S14C). We have included the results in line 245-256, rephrased the discussion in lines 298-302 and included a discussion about MsHi-like in line 302-305.

I don't think the authors are providing any significant findings through the comparison of *M. sexta* Hi with plant His, because properties of MsHi-1 have not been fully investigated. For example, the substrate specificity of MsHi-1 should be examined and the results should be compared with those of plant His.

ANS: Thank you for the suggestion. We now tested MsHi-1 with two additional Z-3-aldehydes; Z-3-octenal and Z-3-nonanal (Fig. S5). The results show that MsHi-1 is able to convert Z-3-aldehydes with different chain lengths as well. This result suggests that MsHi-1 has similar properties as plant Hi's (cupin-like), i.e. the respective proteins do not only re-arrange Z-3-hexenal but also other Z-3 aldehydes (Kunishima M., *et al*, 2016 and Spyropoulou EA., *et al*, 2017). This information has now been added to the result section at lines 137-142, at the discussion at lines 324-326 and in the material method at lines 555-566. As mentioned above, we have now also determined kinetic parameters of MsHi-1 and have added these data in the manuscript in table S2 and at line 123-125 as well as in the material and method section at line 607-619 highlighted in yellow.

The size-exclusion membrane ultrafiltration showed the enzyme should be bigger than 50 kDa, but is it possible that the enzyme consists of subunit structure? If, for example, the enzyme would be a dimer, then, the monomeric unit should be 25-50 kDa. In the sense, the claim that the monomeric structure of the enzyme should be bigger than 50 kDa is inappropriate, at least, before identification of the gene. It would be helpful to estimate molecular weight of the enzyme with the chromatogram obtained with Sephacryl S-300.

ANS: As we did not run any calibration standards with differing molecular weights within the working range of the column, we can indeed not make the claim already early in the manuscript, i.e. before identification of the gene, that the monomeric structure of the enzyme should be bigger than 50 kDa. We have removed this claim from the second result section.

Reviewer #2 (Remarks to the Author):

Key results

In the paper 'A salivary GMC oxidoreductase of *Manduca sexta* re-arranges the green leaf volatile profile of its host plant' a (3Z):(2E)-hexenal isomerase (Hi-1) was identified from the salivary glands of *M. sexta*. This protein not only alters the GLV profiles from host plants, but also is involved in interesting physiological ramifications for the insect during development. The authors also found Hi-1 homologs in other lepidopterans that could catalyze similar reactions.

The purification and identification of the previously described (3Z):(2E)-hexenal isomerase as belonging to the GMC β subfamily represents a significant addition to the field of insect oral secretions and plant GLV emission. The further discovery that Hi-1 knockout mutant adults experienced higher amounts of morphological deformations such as proboscis misfolding, loss of thoracic hair, and non-expanded wings, demonstrates that Hi-1 has an essential role in insect physiological development and the observed function in catalyzing GLV isomers is likely an indirect plant response to this protein.

ANS: Thank you, we are pleased to see that Reviewer 2 saw novelty in our work with a broad impact for future studies.

Data Interpretation

Overall, the validity and robustness of the data interpretations and conclusions are strong and do not over-reach. The reference to the 'change in volatile signal is bittersweet for the insect as it can be used by their natural enemies as a prey location cue' and the referral to this insect-derived enzyme as a 'double-edged sword' (lines and 24-25 and 86, respectively) could benefit from further discussion. Particularly, in light of the findings presented here that concern the role of this protein in insect physiological development, the paper would benefit from discussion of the placement of this protein in the context of the plant-insect 'arms race'.

ANS: Thank you for this suggestion. We agree that the placement of the hexenal isomerase in the context of the plant-insect 'arms-race' would benefit the paper. We have discussed this now in lines 351-368 highlighted in blue.

Context of Results

The results of this study have been provided with sufficient context and consideration of previous work, as reflected in the introduction, discussion, and references.

The focus of this study was to identify the protein involved in the previously reported shift in the emission of GLV isomers, not to perfectly replicate a natural 'real world' scenario with herbivory on whole plants. However, normalizing Hi-1 activity by protein content (lines 288-299), does not account for physiologically or ecologically relevant protein concentrations present in caterpillar saliva or those necessary to isomerize (3Z):(2E)-hexenal in whole plants. Hi-1 activity and the concentration of this protein likely vary among caterpillar species. The determination and use of biologically relevant concentrations of Hi-1 from *B. mori* (and other

caterpillar species) would clarify any differences in results between this study and those found in the literature.

ANS: Thank you for your suggestion. To verify whether OS of *B. mori* indeed contains a protein with Hi activity we produced recombinant protein from one of the putative Bombyx Hi, BGIBMGA000158. We refer to our above responses that describe our validation of Hi activity of recombinant BGIBMGA000158. In short, Recombinant protein of BmHi (Fig. S14A&B) was able to convert Z3AL to E2AL (Fig. S14C) albeit apparently to a lesser extent than MsHi-1 (Fig. 2C). We have included the new results at line 245-256 and rewritten the discussion at lines 298-302.

Methods

This is an extensive study which utilized an impressive suite of techniques and methods to identify Hi-1 and address the role of this protein in isomerizing GLVs from host plants and in insect development, through the use of CRISPR-Cas9, cloning, phylogenetic analysis, VOC collection and analysis, and expression of Hi-1 in two host plants. The data and methods are presented in clear detail.

ANS: Thank you for your positive comments.

Considering the physiological defects in the CRISPR-Cas9 insect mutants (proboscis misfolding, loss of thoracic hair, and non-expanded wings) likely have a negative effect on insect reproduction and survival, it is impressive that these mutant lines could be maintained. Could further clarification be included to account for the maintenance of the CRISPR-Cas9 mutants lines for three generations?

ANS: Thank you for acknowledging the tremendous work effort of our team to maintain the mutants. The mutant colony was indeed not easy to propagate due to their developmental defects (and by now we have lost the colony due to a severe virus infection). As you may see in Fig 7F, approx. 20-30% mutant adults did not show any deformations, hence we were still able to maintain the mutant colony by drastically increasing the population size.

Since GLVs have been successfully collected on adsorption polymers (such as Super Q or Hayesep Q) with dynamic push-pull techniques from both homogenized leaf tissue and whole plants (see: Engelberth J, Engelberth M. Variability in the capacity to produce damage-induced aldehyde green leaf volatiles among different plant species provides novel insights into biosynthetic diversity. *Plants*. 2020 9(2):213. and Allmann S, Baldwin IT. Insects betray themselves in nature to predators by rapid isomerization of green leaf volatiles. *Science*. 2010, ;329(5995):1075-8. among others), it would be beneficial to provide the rationale for using SPME to collect GLVs since VOC collection methodologies have different strengths and limitations. Furthermore, analysis of GLVs by GC-ToF-MS and GC-qToF-MS is less common in the literature than analysis by GC-FID/GC-MS. Why was this analytical method selected for volatile analysis?

ANS: We agree that there are plenty of different methods and machines that can be used to measure volatiles. The probably most obvious reason for choosing one machine over the other

is simply the availability of instruments. GC-ToF-MS as well as GC-qToF-MS are both suitable for measuring volatiles and we have previously used them for similar experiments (e.g.: Allmann S & Baldwin IT. 2010, Science. DOI: 10.1126/science.1191634; Spyropoulou EA *et. al.*, 2017, Frontiers in Plant Science. DOI: 10.3389/fpls.2017.01342)

Concerning the use of SPME, we agree that SPME is not necessarily a quantitative analytical method since only a portion of the compounds might partition into the headspace, and competition effects between volatile compounds can cause biases in the quantitative determination of compounds. However, concentrations of Z-3-hexenal that were used for the assays were low, and clearly within the linear range of detection by SPME. Competition effects on the fiber at these concentrations are thus rather unlikely.

Furthermore, in our case we are only interested in relative numbers and not necessarily in absolute numbers, as we are calculating the percentage/relative amount of E-2-hexenal per reaction: To determine the conversion from Z-3-hexenal to E-2-hexenal we first calculated the sum of aldehydes (Z-3-hexenal + E-2-hexenal) measured by SPME, taking the response factors of each compound into account. We subsequently calculated the percentage of E-2-hexenal and subtracted the non-enzymatic conversion from this value.

Suggested improvements

The described leaf disc assay (lines 602-608) undoubtedly allows for highly controlled volatile collection from treated plant tissue. However, this extensive study would greatly benefit from testing the GLV emission from whole plants treated Hi-1 and OS from the various caterpillar species. Such an addition, as well as the use of biologically relevant Hi-1 concentrations from *M. sexta* saliva would provide a pertinent aspect to this study.

ANS: We appreciate the reviewer's suggestion and agree that testing the GLV emission from whole plants treated with Hi-1 and OS from various caterpillar species, as well as using more biologically relevant Hi-1 concentrations from *M. sexta* saliva, would provide valuable insight into the complex role of Hi. In an independent series of experiments, we have identified and characterized Hi homologs in several lepidopteran species. Here, we indeed applied recombinant protein of the different Hi candidates on wounded leaves of the corresponding host plants, and we were able to identify changes in their GLV-profile. As these challenging experiments are clearly a study on their own, we strongly feel that these are beyond the scope of the current manuscript (which is already heavy in content, e.g. see high number of main and supplemental figures) and we will prepare an extensive follow-up manuscript.

Line 156, 158 and elsewhere: specify which of the two types of caterpillar salivary glands were dissected, mandibular or labial.

ANS: Thanks for pointing this out. We have specified them now as labial glands throughout the manuscript, and all changes are highlighted in blue.

REVIEWERS' COMMENTS

Reviewer #1 (Remarks to the Author):

Reviewer 2's opinion is very pertinent: Hi was not created by the insects to alter GLV composition, but because it is essential for the metabolism of other compounds, which in turn, unfortunately for the insects, alter GLV composition. Perhaps the changes in the GLV composition were not maladaptive to the insects at the beginning, but in the arms race between herbivores and predators, the predators might acquire the ability to recognize the composition in order to find the larvae of *M. sexta* specifically. By properly asserting this, the importance of this paper is now well communicated. The reviewer's comments were appropriately addressed, by showing the kinetics parameter of the recombinant enzyme and adding a discussion of the possibility of secretion of a protein without the secretory signal. Although I still believe that a bioassay using hi-1 KO insects with plant leaves and predators is essential to make the authors' proposal unequivocally, I also understand the situation explained in the responses to reviewers' comments. I look forward to future challenges.

Reviewer #2 (Remarks to the Author):

The authors of the manuscript 'A salivary GMC oxidoreductase of *Manduca sexta* re-arranges the green leaf volatile profile of its host plant' have thoroughly addressed the concerns of this reviewer in the revisions. These revisions added clarity to the project and context as to how this research fits into the overall scope of insect elicitors and effectors and their role in insect fitness and impact on the host plant. The use of labial caterpillar salivary glands was clarified throughout the manuscript. Furthermore, rewrites of the discussion and the inclusion of additional results and supplemental material greatly strengthens the research presented in this manuscript.

Reviewers' Comments

Reviewer #1 (Remarks to the Author):

Reviewer 2's opinion is very pertinent: Hi was not created by the insects to alter GLV composition, but because it is essential for the metabolism of other compounds, which in turn, unfortunately for the insects, alter GLV composition. Perhaps the changes in the GLV composition were not maladaptive to the insects at the beginning, but in the arms race between herbivores and predators, the predators might acquire the ability to recognize the composition in order to find the larvae of *M. sexta* specifically. By properly asserting this, the importance of this paper is now well communicated. The reviewer's comments were appropriately addressed, by showing the kinetics parameter of the recombinant enzyme and adding a discussion of the possibility of secretion of a protein without the secretory signal. Although I still believe that a bioassay using hi-1 KO insects with plant leaves and predators is essential to make the authors' proposal unequivocally, I also understand the situation explained in the responses to reviewers' comments. I look forward to future challenges.

We appreciate all the comments and suggestions from reviewer #1.

Reviewer #2 (Remarks to the Author):

The authors of the manuscript 'A salivary GMC oxidoreductase of *Manduca sexta* re-arranges the green leaf volatile profile of its host plant' have thoroughly addressed the concerns of this reviewer in the revisions. These revisions added clarity to the project and context as to how this research fits into the overall scope of insect elicitors and effectors and their role in insect fitness and impact on the host plant. The use of labial caterpillar salivary glands was clarified throughout the manuscript. Furthermore, rewrites of the discussion and the inclusion of additional results and supplemental material greatly strengthens the research presented in this manuscript.

We thank reviewer #2 for all the comments and suggestions.